# TOKENIZING LOOPS OF ANTIBODIES

## ABSTRACT

The complementarity-determining regions (CDRs) of antibodies are loop structures that are key to their interactions with antigens, and of high importance to the design of novel biologics. Existing approaches for characterizing the diversity of CDRs have limited coverage and cannot be readily incorporated into protein foundation models. Here we introduce **I**mmuno**G**lobulin **LOO**p Tokenizer, **IGLOO**, a multimodal antibody loop tokenizer that encodes backbone dihedral angles and sequence. IGLOO is trained using a contrastive learning objective to map loops with similar backbone dihedral angles closer together in latent space. Compared to state-of-the-art protein encoding approaches, IGLOO can efficiently retrieve the closest matching loop structures from a structural antibody database, outperforming the existing methods on identifying similar H3 loops by 6.1%. IGLOO assigns tokens to all loops, addressing the limited coverage issue of canonical clusters, while retaining the ability to recover canonical loop conformations. To demonstrate the versatility of IGLOO tokens, we show that they can be incorporated into protein language models with IGLOOLM and IGLOOALM. On predicting binding affinity of heavy chain variants, IGLOOLM outperforms the base protein language model on 8 out of 10 antibody-antigen targets. Additionally, it is on par with existing state-of-the-art sequence-based and multimodal protein language models, performing comparably to models with $7\times$ more parameters. IGLOOALM samples antibody loops which are diverse in sequence and more consistent in structure than state-of-the-art antibody inverse folding models. We show that IGLOO can rapidly and scalably prioritize functional antibody variants from large mutagenesis libraries, achieving a $1.9\times$ enrichment of experimentally validated HER2 binders in a zero-shot setting. IGLOO demonstrates the benefit of introducing multimodal tokens for antibody loops for encoding their diverse landscape, improving protein foundation models, and for antibody CDR design.

## 1 INTRODUCTION

Antibodies are a class of proteins that are essential in the body's immune response and a widely used therapeutic modality (Crescioli et al., 2025). They are comprised of two identical light and two identical heavy chains. The light and heavy chains are divided into a constant and variable domain, where the variable domain is comprised of complementarity-determining regions (CDRs),[1] which are structurally distinct flexible loops between antiparallel beta strands in the immunoglobulin fold. The CDRs play an essential role in the antibody's ability to recognize and bind antigens in a highly specific manner (Xu & Davis, 2000). Protein and antibody language models trained on amino acid sequence tokens have been powerful for learning evolutionary patterns that are useful for function prediction (Kulmanov et al., 2024), sequence design (Zhao et al., 2025), and variant effect prediction (Hie et al., 2024; Notin et al., 2023).

Recently, the development of multimodal protein language models (Su et al., 2023; Heinzinger et al., 2024; Hayes et al., 2025) has expanded to incorporate structure tokens in addition to sequence tokens. However, such approaches tokenize structures at the amino acid level, focus on reconstruction, and do not consider the higher-level organization and modularity of protein domains (Sigrist et al., 2010; Mistry et al., 2021). A multimodal tokenizer for antibodies should therefore consider the inherent organization in antibody structures and sequences for effective representation learning.

---

[1]Here we consider four CDR regions. The fourth CDR is the loop joining the D and E strands adjacent to CDR1 and CDR2, which is often considered part of the framework (Kelow et al., 2020).

Tokenization or clustering of immunoglobulin loop regions based on their dihedral backbone angles into 'canonical clusters' has been adopted since Chothia & Lesk (1987). Such a grouping of loops has been useful for understanding the structural diversity of antibodies (Teplyakov et al., 2016), designing antibody loops with consistent structure and diverse sequences (Adolf-Bryfogle et al., 2018), and for studying conformational changes of antibody loops in molecular dynamics simulations (Fernández-Quintero et al., 2020; 2019). These approaches are limited by (1) their limited coverage of antibody structures. Recent clusters defined by Kelow et al. (2022) reveal that 20.3% of loops do not map to any known canonical cluster, with the proportion increasing to 76.3% for H3 loops (Table S6). (2) All existing clusters only consider backbone coordinates or dihedral angles, without incorporating sequence information. Finally, (3) existing clusters cannot be readily applied to protein language models. Thus, the tokenization of immunoglobulin loops for multimodal representation remains an open challenge.

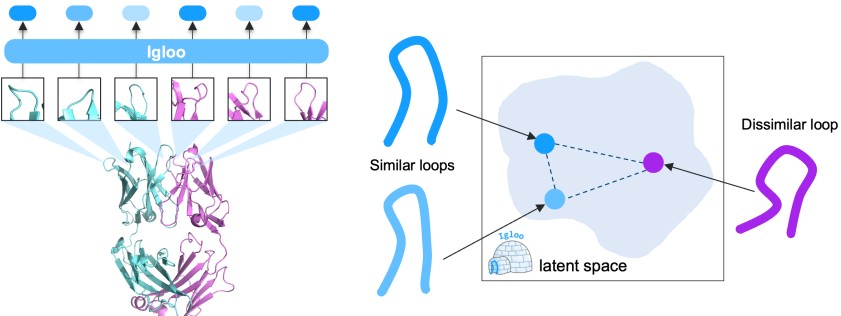

Figure 1: **Left** IGLOO is a multimodal tokenizer for antibody loops. **Right** Organization of the IGLOO latent space is achieved through a contrastive learning objective on dihedral angle distance between backbones.

**Present work.** We introduce **I**mmuno**G**lobulin **LOO**p Tokenizer, **IGLOO**, a *multimodal antibody loop tokenizer* for encoding backbone dihedral angles and sequence (Fig. 1). Unlike existing structure tokenizers, which focus on the amino acid scale, IGLOO tokenizes at the substructure loop level. IGLOO is trained on 807,815 loop regions from heavy and light chains of experimentally-derived and computationally predicted structures. We develop a contrastive learning objective based on the dihedral angle distance defined by North et al. (2011) to train IGLOO. While being a self-supervised model, IGLOO successfully reproduces known canonical conformations assigned for 90.6% of loops in SAbDab Dunbar et al. (2014). To demonstrate the versatility and utility of IGLOO tokens, we present four key applications:

- **Retrieval of similar loop structures from large structural databases.** By learning to compare loop structures, IGLOO retrieves more similar loop structures from SAbDab compared to state-of-the-art protein encoding approaches. For the H3 loop, which exhibits the most sequence and structure diversity, IGLOO outperforms the previous best model on retrieving H3 loops with similar dihedral angle backbones by 6.1%.

- **Improved antibody affinity prediction with protein language models.** We incorporate IGLOO loop tokens into an antibody language model, and train IGLOOLM. By using the representations learned from IGLOOLM to predict binding affinity of heavy chain variants, we show it outperforms the base model on 8 out of 10 antibody-antigen targets and performs on par to models with $7\times$ more parameters.

- **Sampling diverse loops with consistent structure.** IGLOOALM is a protein language model with the IGLOO loop tokens and IGLOO multimodal residue tokens. When loop sequences are masked out, the loops sampled from IGLOOALM are diverse in sequence and more consistent in structure than state-of-the-art antibody inverse folding models. Redesigned CDR H3 loops of a SARS-CoV-2 antibody with IGLOOALM achieves an average sequence identity of 0.27 while achieving less than 1Å RMSD to the original loop.

- **Prioritization of loops for library design.** We demonstrate that IGLOO enables zero-shot, highly scalable hit-to-lead optimization by prioritizing experimentally-validated CDR H3 loop variants for the HER2 target. Sequences sharing the same IGLOO token as the seed

loop exhibit 55.3% experimentally validated binders, a $1.9\times$ enrichment over the baseline library.

By introducing multimodal tokens for antibody loops, IGLOO captures the structural and functional diversity of loop conformations, improving the expressiveness of protein foundation models and advancing rational antibody design.

## 2 RELATED WORK

**Tokenization of protein structure.** The construction of classifications of protein structures at the domain level has been applied for understanding the relationship with domain function (Lo Conte et al., 2000; Ouzounis et al., 2003; Sigrist et al., 2010; Mistry et al., 2021). Learning structurally informed residue-level representations can be achieved with geometric features (Jing et al., 2020), multiview contrastive learning between sequence and structure views of the same protein (Zhang et al., 2022), hierarchical graph neural network on the protein structure (Wang et al., 2022), and with intermolecular interactions (Fang et al., 2025). Tokenization at the amino acid level has shown significant advances in the speed of protein structure search with Foldseek (Van Kempen et al., 2024). Yuan et al. (2025) compare different approaches for tokenizing amino acid structures including VQVAE (Hayes et al., 2025) and inverse-folding-based methods (Dauparas et al., 2022).

**Multimodal protein language models.** Multimodal protein language models have been trained to learn meaningful representations and to generate over sequence, structure, and function. Models such as SaProt (Su et al., 2023) and ProstT5 (Heinzinger et al., 2024) learn representations from protein sequence and Foldseek 3Di amino acid tokens, which capture structural information. ProSST (Li et al., 2024) represents proteins with sequence and residue-level structure tokens that capture local environments. ProSSN (Tan et al., 2025) uses both sequence and the topological structure of proteins to learn multimodal representations. ESM3 (Hayes et al., 2025) is a generative model that models the sequence, structure, and function of amino acids simultaneously.

**Clustering Immunoglobulin Loops.** The CDRs of antibodies demonstrate the most variability and are essential to the binding of antibodies to antigens. Thus, there has been significant effort in categorizing all known structures of CDRs (Chothia & Lesk, 1987; Shirai et al., 1996; North et al., 2011; Adolf-Bryfogle et al., 2015; Nowak et al., 2016; Wong et al., 2019b; Kelow et al., 2022; Liu et al., 2024). CDRs fold into a loop structure, and a pair of loops can be compared through their backbone dihedral angles (North et al., 2011). While most approaches only cluster loops of the same length, Nowak et al. (2016) explore clustering of loops of different lengths by aligning loops with their stem region (subsequence of amino acids before and after the loop region) and comparing the resultant RMSD between the loops. SCALOP (Wong et al., 2019a) predicts canonical loops from protein sequences for large-scale annotation of antibody libraries. Zhang et al. (2025) train their model to learn the RMSD between pairs of loops and show how it can be used for designing CDRs. Current methods are limited as many CDRs, especially H3 loops, do not have known canonical conformations (Table S6). We extend existing approaches through the self-supervised definition of antibody loop clusters.

## 3 METHOD

IGLOO is a multimodal tokenizer that incorporates both sequence and backbone structure of the loop structures. Here we focus on modeling loops within antibodies and TCRs, which are the four CDRs of the heavy and light chains. IGLOO is a *tokenizing function* that maps for a loop sequence and backbone structure, to a token $\mathbf{t}$.

**Problem definition.** An antibody loop with $n$ residues is defined by: (1) a sequence of amino acids $\mathbf{a} = (a_1, \ldots, a_n)$ where $\forall i$, $a_i \in \mathcal{V} = \{\text{Ala}, \text{Arg}, \ldots, \text{Tyr}, \text{Val}\}$, which are canonical amino acid residues, and (2) their backbone dihedral angles $\boldsymbol{\phi}, \boldsymbol{\psi}, \boldsymbol{\omega} \in (-\pi, \pi]^n$ (Fig. 2). Our goal is to train a tokenizer $f(\cdot)$ for antibody loops such that $f(\mathbf{a}, \boldsymbol{\phi}, \boldsymbol{\psi}, \boldsymbol{\omega}) = \mathbf{t}$, where $\mathbf{t} \in \mathbb{R}^d$ and $d$ is the token embedding dimension. The token $\mathbf{t}$ supports (1) retrieval of structurally similar loops, (2) integration into protein language models, and (3) guided loop generation.

## 3.1 MULTIMODAL TOKENIZATION OF LOOPS

The input to IGLOO is a loop of length $n$ with dihedral angles $(\phi, \omega, \psi) \in (-\pi, \pi]^{n \times 3}$ and amino acid identities $\mathbf{a}$. The dihedral angles are first converted into coordinates on the unit circle $(\cos \phi, \sin \phi, \cos \psi, \sin \psi, \cos \omega, \sin \omega) \in [-1, 1]^{n \times 6}$ and then projected with a linear layer $\mathbf{D} = (\cos \phi, \sin \phi, \cos \psi, \sin \psi, \cos \omega, \sin \omega) \mathbf{W}_{\text{dihedral}}^T + \mathbf{b}_{\text{dihedral}}$,

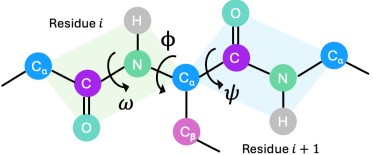

Figure 2: Backbone dihedral angles for residue $i$.

where $\mathbf{D} \in \mathbb{R}^{n \times d}$, $\mathbf{W}_{\text{dihedral}} \in \mathbb{R}^{d \times 6}$, and $\mathbf{b}_{\text{dihedral}} \in \mathbb{R}^d$. For the loop sequence, it is encoded with 20 learnable embeddings for each of the canonical amino acid types into $\mathbf{A} \in \mathbb{R}^{n \times d}$. Next, we sum the sequence and dihedral angle embeddings to produce a multimodal embedding, $\mathbf{X} = \mathbf{D} + \mathbf{A} \in \mathbb{R}^{n \times d}$ (Fig. 3a). To learn a representation across the loop residues $\mathbf{X} = (\mathbf{x}_1, \dots, \mathbf{x}_n)$, we use a transformer architecture based on BERT (Devlin et al., 2019) using the ESM-2 implementation (Lin et al., 2023). A learnable classification token, $\mathbf{t}$, is added to the start of each sequence to learn a meaningful overall representation of the loop.

## 3.2 IGLOO SELF-SUPERVISED TRAINING OBJECTIVES

We train IGLOO with three objectives (Fig. 3b): (1) multimodal masking with reconstruction of dihedral angles and amino acid identities, (2) contrastive learning of protein backbones, and (3) codebook learning.

**Multimodal masking.** Following the multimodal masking approach of ESM-3 (Hayes et al., 2025), we randomly mask 30% of positions per loop. For the loops in the training dataset, we use the following regimes: (1) mask 30% of sequence and dihedral angles for 20% of loops during training, (2) mask 30% of dihedral angles only for 20% of loops during training, (3) mask 30% of sequence only for 20% of loops during training, (4) fully mask the sequence for 10% of loops during training, (5) fully mask the dihedral angles for 10% of loops during training, and (6) no masking for 20% of loops during training.

For the reconstruction of dihedral angle $\theta_i$ of residue $i$ in a loop, the predicted unit circle coordinates $(x_i, y_i)$ are given by passing the hidden representation of the residue to a two-layer MLP. The reconstruction loss $\ell_{\text{dihedral recon.} i}$ is given by the mean squared error between $(\cos \theta_i, \sin \theta_i)$ and $(\cos \hat{\theta}_i, \sin \hat{\theta}_i)$, where $\hat{\theta}_i = \text{atan2}(y_i, x_i)$. We also add a penalty term, $\ell_{\text{dihedral reg.}}$, to regularize the model for the reconstruction of coordinates on the unit circle (Pavllo et al., 2018).

For the prediction of the masked amino acids, the logits of the masked amino acids are given by passing the hidden representation of the residue to a two-layer MLP. The amino acid masking loss for amino acid $i$ in a loop, $\ell_{\text{AA},i}$, is given by $\ell_{\text{AA},i} = -\sum_{a=1}^{20} y_{i,a} \log \hat{p}_{i,a}$, where the predicted probability that the identity is amino acid $a$ is $\hat{p}_{i,a}$, and $y_{i,a}$ is a one-hot indicator of whether the amino acid is of identity $a$.

**Contrastive learning of protein backbones.** For IGLOO to learn a token, $\mathbf{t}$, such that similar tokens share similar loop conformations, we define a contrastive loss function. North et al. (2011) define the similarity between two loops $u$ and $v$ of length $n_u$ and $n_v$, respectively, and $n_u \leq n_v$. The loops have dihedral angles $(\phi^u, \psi^u, \omega^u) \in (-\pi, \pi]^{n_u \times 3}$ and $(\phi^v, \psi^v, \omega^v) \in (-\pi, \pi]^{n_v \times 3}$. The dihedral distance $\mathcal{D}$ is defined as

$$\mathcal{D} = \frac{1}{3n_u} \sum_{\theta \in \{\phi, \psi, \omega\}} \sum_{i=1}^{n_u} 2(1 - \cos(\theta_i^u - \mathcal{P}(\theta_i^v))), \tag{1}$$

where $\mathcal{P}$ aligns residues of loop $v$ to residues of loop $u$. When $n_u = n_v$, the alignment is a one-to-one mapping between the residues. Otherwise, we define an alignment between the two loops using a dynamic time warping path (see Appendix D).

Proteins are chiral molecules, and the orientation of the backbone frame has a strong influence on the atomic structure through side-chain positioning. Prominent examples exist where the backbone RMSD of a pair of loops is low, yet dihedral angles can be up to $180°$ apart with opposite-pointing side chains (North et al., 2011). Therefore, we use dihedral angle distance $\mathcal{D}$ over RMSD to capture the nuances of the loop structure.

A pair of loops $u, v$ is a positive pair ($Y_{uv} = 1$) if the loops are of the same length and $\mathcal{D} < 0.1$. A pair of loops is a negative pair ($Y_{uv} = 0$) if they are of different lengths or $\mathcal{D} > 0.47$ for loops of the same length, where $\mathcal{D} = 0.47$ corresponds to an average difference in dihedral angles of $40°$, which is the threshold used in the clustering by Kelow et al. (2022). Otherwise, the pair of loops is ignored. The *dihedral loss* is the mean binary cross-entropy over the pairs of loops in the batch with positive and negative labels.

$$\ell_{\text{contrastive},uv} \; = \; \text{BCE}\Big(\sigma\Big(\frac{\mathbf{h}_u^\top \mathbf{h}_v}{\tau}\Big), Y_{uv}\Big), \tag{2}$$

where $\mathbf{h}_u = \frac{\mathbf{t}_u}{\|\mathbf{t}_u\|_2}$, $\mathbf{t}_u$ is the classification token embedding for loop $u$ in the batch, and $\tau$ is the temperature. We apply contrastive learning instead of predicting $\mathcal{D}$ as the pretraining task for similar loops to be close in the latent space. A margin between positive and negative pairs is applied so IGLOO does not overfit its representations to the threshold used for the definition of canonical clusters. The IGLOO model presented here only has positive pairs between loops of the same length ($n_u = n_v$). We explore a variant of the model with positive pairs defined for mismatched loop lengths in Appendix D.

**Codebook learning.** In addition to learning continuous tokens, the assignment of loops to $K$ discrete tokens offers a convenient and fast approach for loop comparison required for high-throughput queries. For the codebook $\mathbf{C} \in \mathbb{R}^{K \times d}$ to learn quantized tokens $\hat{\mathbf{t}}$, we include a codebook learning loss (Van Den Oord et al., 2017) on the classification token of loop $u$ with $\ell_{\text{codebook},u} = \|\operatorname{sg}[\mathbf{t}_u] - \hat{\mathbf{t}}_u\|_2^2 + \alpha\|\mathbf{t}_u - \operatorname{sg}[\hat{\mathbf{t}}_u]\|_2^2$, where $\operatorname{sg}[\cdot]$ is the stop gradient operator and $\alpha$ is the weight on the second commitment loss term.

### 3.3 TRAINING AND INFERENCE OF IGLOO

For the training of IGLOO we use the overall loss function, which is given by

$$\mathcal{L} = \ell_{\text{dihedral recon.}} + \ell_{\text{AA}} + \ell_{\text{contrastive}} + \ell_{\text{codebook}} + \lambda\ell_{\text{dihedral reg.}}. \tag{3}$$

We train IGLOO on heavy and light chain CDR1, CDR2, CDR3, and CDR4 loops from all antibodies and nanobodies in SAbDab (Dunbar et al., 2014), and TCRs in STCRDab (Leem et al., 2018). In addition, we also train with Ibex (Dreyer et al., 2025) predicted structures of paired heavy and light chain antibodies from paired sequences of the Observed Antibody Space (OAS, Olsen et al. (2022)) (Appendix A.1). For each antibody, we then use their concatenated CDR sequence and an 80% sequence identity threshold for splitting loops of antibodies into train, test, and validation. Since clustering at the level of concatenated sequences of CDRs can still result in an individual CDR sharing the same sequence, any identical training loop sequences are removed from the validation and test set. In total, IGLOO is trained on 108,167 experimentally resolved loop structures from SAbDab and STCRDab and 699,648 predicted loop structures from paired OAS sequences. At inference, IGLOO outputs a continuous classification loop token $\mathbf{t}$, a quantized token $\hat{\mathbf{t}}$, and a multimodal representation for each residue $i$ in the loop $\mathbf{x}_i$.

### 3.4 INCORPORATING IGLOO TOKENS INTO PROTEIN LANGUAGE MODELS

**Approach.** We demonstrate how IGLOO loop tokens, $\mathbf{t}$, can be inserted as special tokens in protein language models with two complementary approaches. (1) IGLOOLM (Fig. 3c) is a protein language model with the IGLOO loop token, $\mathbf{t}$, inserted at the start of each CDR loop and an `<end>` token added at the end of the loop. (2) IGLOOALM (ALM=dihedral Angle Language Model, Fig. 3d) is a protein language model with the IGLOO loop token, $\mathbf{t}$, and IGLOO multimodal residue tokens, $\mathbf{x}_i$, for each amino acid in the CDR loop. These models are finetuned from the 420M parameter base antibody language model, IgBert (Kenlay et al., 2024), a BERT-style model trained on all paired and unpaired OAS sequences. We project IGLOO tokens with a learnable linear layer so that they are the same dimension as the hidden dimension of tokens in the base protein language model.

Learnable classification tokens have been widely used in text, vision, and single-cell transformers (Devlin et al., 2019; Dosovitskiy et al., 2020; Cui et al., 2024). Analogous to the cell-prompting and gene-prompting paradigms of scGPT (Cui et al., 2024), IGLOOLM encodes loops with tokens, $\mathbf{t}$, while IGLOOALM extends this scheme by combining loop tokens $\mathbf{t}$ and multimodal residue tokens,

**x.** Embeddings from IGLOOLM contain the context of the loop conformation, while embeddings from IGLOOALM additionally contain the context of the dihedral angles of each residue in the loop. We demonstrate how IGLOOALM excels in tasks where the accurate residue-level structure is provided. Conversely, IGLOOLM excels in tasks where the loop conformation is known, but accurate residue-level structure prediction is challenging – for example, deep-mutational-scan datasets in which sequences differ by only a few point mutations (Pak et al., 2023; Buel & Walters, 2022).

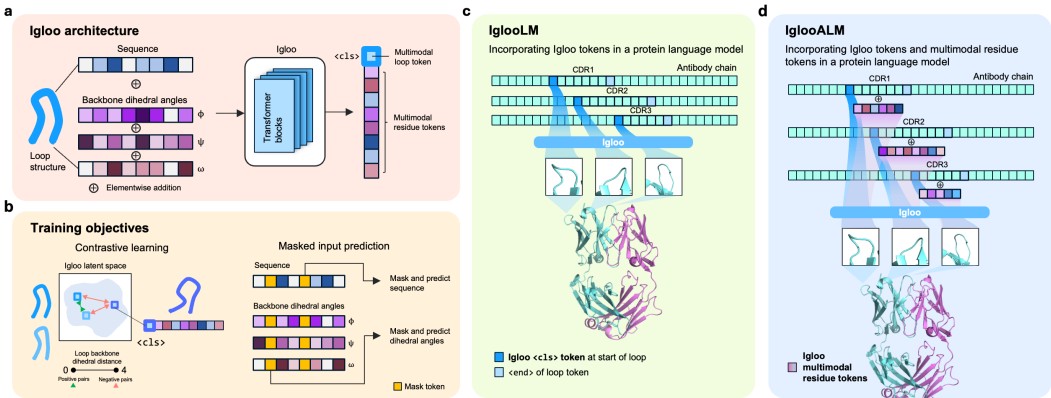

Figure 3: **a** IGLOO is a multimodal tokenizer for antibody loops. **b** Training objectives involve (1) contrastive learning with positive and negative pairs defined by their dihedral angle distance, and (2) masking and prediction of sequence and backbone dihedral angles. IGLOO tokens of CDR loops can be incorporated into protein language models where **c** IGLOOLM contains only the `<cls>` (classification) loop token, **t**, and **d** IGLOOALM contains the loop token and multimodal residue tokens.

**Training and Inference.** The models are finetuned with the same masked language model objective as Kenlay et al. (2024)–uniformly randomly masking 15% of amino acid residues, for which 80% are then replaced by a masked token, 10% are changed to a random token in the vocabulary, and 10% are left unchanged. For the masked tokens in the loop, we keep the dihedral angles, but the sequence information of these residues is also masked for the computation of the IGLOO token. IGLOOLM and IGLOOALM are trained on single domains from paired OAS sequences, which are split into train, validation, and test splits based on a 90% sequence identity split (Appendix A.2). All structures of the loops required for IGLOO tokens are extracted from Ibex predicted structures.

## 4 EXPERIMENTS

### 4.1 IGLOO FOR PARATOPE RETRIEVAL

In this evaluation, for a set of query CDR loop structures, we task IGLOO to retrieve from a large repository of CDRs those with the closest experimentally determined backbone structure, thereby directly assessing how well the representation captures paratope-level structure.

**Experimental setup.** Query CDRs are from the IGLOO unseen test set of CDRs, and the repository CDRs are those from the training and validation set of SAbDab. We use the IGLOO token **t** and retrieve 20 loops with the highest cosine similarity from loops of the same type and length. Retrieved CDRs are deemed correct if $\mathcal{D} < 0.47$ (Eq. 1) or RMSD $< 1$ Å to the query CDR loop.

**Baselines.** We compare IGLOO to protein language models that are trained on UniProt: ESM C (ESM Team, 2024) and ESM-2 (3B) (Lin et al., 2023), and models trained on OAS: AbLang2 (Olsen et al., 2024) and IgBert (Kenlay et al., 2024). Additionally, we evaluate the quality of retrieval compared to multimodal protein language models, SaProt (Su et al., 2023) and ProstT5 (Heinzinger et al., 2024), which also take as input the Foldseek 3Di tokens (Van Kempen et al., 2024) derived from the protein structure. To ensure embeddings focus on the loop, for models which embed the whole protein sequence, the loop embedding is defined as the mean embedding over the amino acids in the loop. We also compare to continuous structure tokenizers that were benchmarked by (Yuan et al., 2025). This includes inverse folding models: MIF (Yang et al., 2023) and ProteinMPNN

(Dauparas et al., 2022), and the continuous encoder embedding of VQVAE models: Foldseek 3Di and Amino Aseed (Yuan et al., 2025). For further details see Appendix B.3.

**Results.** We evaluate the models with Precision@20 on experimentally determined structures of CDRs in our SAbDab test set. IGLOO achieves state-of-the-art performance in retrieving similar paratopes, based on dihedral distance $\mathcal{D} < 0.47$ and RMSD $< 1$ Å from loops of the same length (Table 1). We emphasize $\mathcal{D}$ because RMSD can be low even when dihedral angles flip by $180°$ reversing side-chain orientations. Conversely, $\mathcal{D}$ captures these chirality-sensitive differences. Compared to larger protein language models pretrained with masked language modeling, structure encoders and IGLOO achieve higher Precision@20. For H3 specifically, Precision@20 for RMSD $< 1$ Å is 0.278 for IGLOO compared to 0.292 for Amino Aseed, but with the dihedral distance $\mathcal{D}$ that our training optimizes for, IGLOO is 0.402 compared to 0.379 for Amino Aseed (+6.1%). Ablating the margin in the contrastive loss of IGLOO resulted in Precision@20 for $\mathcal{D}$ to increase to 0.417 (+10.0% relative to Amino Aseed, Table S3), but this ablated model does not uniformly help other CDRs (for further details on model ablations see Appendix C). The H3 loop is particularly hard to represent for sequence-only language models due to the high sequence diversity owing to V(D)J recombination (Tonegawa, 1983). We also trained a version of IGLOO trained with positive pairs defined across different length loops (Appendix D, 'Mismatched length' in Table S3), which yields broadly comparable Precision@k while slightly reducing performance on some loop types.

Table 1: Precision@20 for retrieval of similar CDR paratopes for experimentally determined structures in the SAbDab test set. The **first**, second, and *third* best performance for each column are highlighted. Additional results for precision at rank 1, 5, and 10 are available at Table S5.

| | Model | % RMSD $< 1$ Å | | | | | | % $\mathcal{D} < 0.47$ | | | | | |
|---|---|---|---|---|---|---|---|---|---|---|---|---|---|
| | | L1 | L2 | L3 | H1 | H2 | H3 | L1 | L2 | L3 | H1 | H2 | H3 |
| **Random** | | 0.545 | 0.557 | 0.373 | 0.249 | 0.351 | 0.127 | 0.648 | 0.730 | 0.392 | 0.559 | 0.508 | 0.126 |
| **PLM** | ESM C | 0.750 | 0.700 | 0.489 | 0.418 | 0.519 | 0.190 | 0.811 | 0.916 | 0.517 | 0.692 | 0.702 | 0.208 |
| | ESM-2 (3B) | 0.740 | 0.704 | 0.500 | 0.425 | 0.522 | 0.206 | 0.802 | 0.904 | 0.534 | 0.706 | 0.688 | 0.237 |
| **AbLM** | AbLang2 | 0.689 | 0.604 | 0.482 | 0.402 | 0.497 | 0.173 | 0.761 | 0.782 | 0.537 | 0.602 | 0.699 | 0.222 |
| | IgBert | 0.705 | 0.622 | 0.482 | 0.377 | 0.479 | 0.182 | 0.773 | 0.813 | 0.511 | 0.709 | 0.677 | 0.216 |
| **MPLM** | SaProt | 0.737 | 0.704 | 0.499 | 0.420 | 0.491 | 0.218 | 0.790 | 0.918 | 0.578 | 0.688 | 0.646 | 0.248 |
| | ProstT5 | 0.782 | **0.716** | 0.539 | 0.458 | 0.586 | 0.276 | 0.846 | 0.941 | *0.629* | 0.711 | **0.756** | 0.359 |
| **IF** | MIF | 0.776 | 0.699 | 0.516 | 0.432 | 0.491 | 0.231 | 0.833 | 0.933 | 0.604 | 0.702 | 0.641 | 0.298 |
| | ProteinMPNN | 0.804 | 0.700 | *0.546* | 0.459 | 0.521 | 0.286 | 0.839 | *0.943* | 0.632 | **0.732** | 0.710 | *0.372* |
| **VQVAE** | Foldseek 3Di | 0.785 | 0.696 | 0.556 | **0.467** | **0.591** | *0.281* | 0.849 | 0.909 | 0.640 | 0.715 | *0.730* | 0.362 |
| | Amino Aseed | **0.812** | 0.713 | 0.542 | 0.420 | 0.529 | **0.292** | **0.851** | 0.952 | 0.625 | 0.688 | 0.713 | 0.379 |
| **Ours** | IGLOO | *0.793* | *0.705* | **0.558** | 0.459 | *0.578* | 0.278 | **0.851** | **0.956** | **0.674** | 0.715 | 0.749 | **0.402** |

PLM: Protein Language Model, AbLM: Antibody Language Model,
MPLM: Multimodal Protein Language Model, IF: Inverse Folding Model

## 4.2 IGLOO FOR CLUSTERING ANTIBODY STRUCTURES

The canonical clusters established by North et al. (2011) and Kelow et al. (2022) have been widely used for categorizing new structures (Teplyakov et al., 2016) and to analyze molecular dynamics simulations of antibodies (Fernández-Quintero et al., 2020; 2019). In this section, we evaluate how well the quantized token, $\hat{\mathbf{t}}$, recovers their clusters of antibody CDRs.

**Evaluation setup.** Let the IGLOO learned codebook $\hat{\mathbf{t}}$ induce the partition $\mathcal{C} = \{C_1, \ldots, C_K\}$, and let $\mathcal{G} = \{G_1, \ldots, G_L\}$ denote the reference canonical clusters of Kelow et al. (2022). We quantify the agreement between these two partitions using *cluster purity*. For each predicted cluster $C_k$, we select the dominant reference class based on majority vote: $y^\star(k) = \arg\max_\ell |C_k \cap G_\ell|$. Items in $C_k$ whose reference label equals $y^\star(k)$ are considered correctly assigned. Overall accuracy is the proportion of correctly assigned instances.

$$\text{Purity}(\mathcal{C}, \mathcal{G}) = \frac{1}{N} \sum_{k=1}^{K} \max_\ell |C_k \cap G_\ell|, \quad N = \sum_{k=1}^{K} |C_k|. \tag{4}$$

We evaluate on all loops in SAbDab that can be assigned to a reference cluster with a cutoff of $\mathcal{D} = 0.47$ to the centroid. A limitation of the existing canonical clustering approach is that several loops are not assigned to any cluster. We do not evaluate cluster purity on unassigned loops, which are typically referred to as belonging to "noise" clusters.

The reference definition of clusters assigns different clusters for different loop types and lengths. We also evaluate the cluster's *loop-type purity* and *loop-length purity*, defined as:

$$p_k^{\text{type}} = \frac{1}{n_k} \max_t \sum_{x \in C_k} \mathbb{1}\{\text{loop type}(x) = t\}, \; p_k^{\text{len}} = \frac{1}{n_k} \max_\ell \sum_{x \in C_k} \mathbb{1}\{\text{loop length}(x) = \ell\}, \quad (5)$$

where $t \in \{\text{H1, H2, H3, H4, L1, L2, L3, L4}\}$. We report global scores with a weighted average of the cluster-level purity scores, $P^{\text{type}} = \frac{1}{N} \sum_k n_k p_k^{\text{type}}, P^{\text{len}} = \frac{1}{N} \sum_k n_k p_k^{\text{len}}$, where $n_k$ is the number of loops in the cluster and $N = \sum_k n_k$.

**Results.** Across SAbDab, 1305 IGLOO codebooks and 180 reference clusters are used. Without exposure to loop-type annotations, the IGLOO-induced partition is highly homogeneous, attaining a loop-type purity of $P^{\text{type}} = 0.983$, and loop length purity $P^{\text{len}} = 0.965$. Visualization of the latent space in 2D with UMAP also shows localization of loops by loop type, length, and canonical cluster (Fig. S1). We report cluster purity in Table S7. Our results are comparable with Wong et al. (2019a), which uses Position-Specific Scoring Matrices to predict canonical forms from sequence. These results highlight that IGLOO can recover the known canonical clusters with high purity.

We further explore how different loops differ in their distribution across codebooks. The proportion of each loop in SAbDab assigned to the top 20 used codebooks is shown in Fig. S2 for each loop type. The H4 and L2 loop types have relatively low diversity with 93.0% and 91.7% of loops assigned to a codebook in the top 20, respectively. As expected, the most diverse loop, H3, has the lowest coverage in the top 20 codebooks with 14.6% of loops. The most frequent H3-loop codebook entry appears 387 times. Every occurrence shares an identical loop sequence derived from single-chain Fv16 antibody structures, a scaffold that is widely represented in the PDB.

## 4.3 PREDICTING BINDING AFFINITY WITH IGLOOLM

Next, IGLOOLM is evaluated on datasets where sequences differ by a few variants. For a set of heavy chain antibody mutants, we apply the protein-level representations of the heavy chain sequences to predict binding affinity. This section aims to test if incorporating IGLOO tokens, $t$, into protein language models as a special token is beneficial to the representations learned by the model.

**Experimental setup.** We use the curated set of antibody-antigen binding affinity dataset from AbBiBench (Zhao et al., 2025). Antibody structures for the variants are predicted with Ibex, used as input to obtain IGLOO tokens, and embeddings are then generated with IGLOOLM. Sequence-level embeddings are obtained by averaging residue-level embeddings. A separate model is trained for each antibody-antigen pair. Sequence-level embeddings are used as input to train a ridge regressor evaluated with 10-fold nested cross-validation (Appendix B.4). Models are evaluated with the Spearman correlation coefficient, $\rho$, between the predicted and true binding affinity.

**Baselines.** The protein language models, ESM C, ESM-2 (3B), AbLang2, and IgBert, and multimodal protein language models SaProt and ProstT5 introduced in Section 4.1 are used as baselines. We obtain sequence-level embeddings by averaging the residue-level embeddings. For the multimodal protein language models, we use Foldseek 3Di tokens from the Ibex predicted structures.

**Results.** Across the 10 antibody–antigen pairs in Table 2, IGLOOLM surpasses the base model IgBert from which it is derived on 8 cases. It ranks first or second on 7 of the 10 pairs. Structure is not always beneficial on AbBiBench: ProstT5 and SaProt underperform sequence-only protein language models, and incorporating residue-level dihedral angle tokens with IGLOOALM also underperforms compared to IGLOOLM (Table S8). We attribute this to residue-level noise in predicted loop structures differing by a few variants. Therefore, showing the benefit of incorporating only the IGLOO classification token into IGLOOLM for representing loops which differ by a few mutations. As protein language models improve with scale (Lin et al., 2023), it is notable that IGLOOLM, a 420M parameter model which is more than 7× smaller than ESM-2 (3B), achieves better performance on average across the 10 antibody-antigens.

Table 2: Spearman correlation coefficient (↑) for binding affinity prediction across 10 different targets from AbBiBench. The **first** and second values are highlighted. We report the standard error across the 10 fold cross-validation in parentheses.

| Target | ESM C (300M) | ESM-2 (3B) | SaProt (1.3B) | ProstT5 (3B) | AbLang2 45M | IgBert (420M) | IGLOOLM (420M) |
|---|---|---|---|---|---|---|---|
| **1mlc** | 0.609 (0.017) | 0.551 (0.013) | 0.557 (0.020) | 0.280 (0.040) | 0.634 (0.015) | **0.665 (0.015)** | 0.616 (0.009) |
| **1n8z** | 0.673 (0.022) | 0.635 (0.019) | 0.637 (0.028) | 0.351 (0.057) | 0.646 (0.021) | **0.682 (0.023)** | 0.675 (0.025) |
| **2fjg** | **0.809 (0.010)** | 0.752 (0.010) | 0.754 (0.012) | 0.355 (0.021) | 0.752 (0.007) | 0.694 (0.013) | 0.713 (0.014) |
| **3gbn_h1** | 0.901 (0.004) | **0.953 (0.003)** | 0.915 (0.005) | 0.638 (0.013) | 0.945 (0.004) | 0.947 (0.004) | 0.948 (0.004) |
| **3gbn_h9** | 0.932 (0.004) | **0.971 (0.002)** | 0.952 (0.003) | 0.679 (0.017) | 0.963 (0.003) | 0.961 (0.003) | 0.962 (0.003) |
| **4fqi_h1** | 0.871 (0.001) | **0.955 (0.001)** | 0.866 (0.001) | 0.593 (0.002) | 0.883 (0.001) | 0.898 (0.001) | 0.921 (0.001) |
| **4fqi_h3** | 0.936 (0.001) | **0.973 (0.001)** | 0.958 (0.001) | 0.644 (0.009) | 0.969 (0.001) | 0.970 (0.001) | 0.971 (0.001) |
| **aayl49** | 0.617 (0.010) | 0.584 (0.013) | 0.584 (0.012) | 0.301 (0.014) | 0.563 (0.010) | 0.611 (0.010) | **0.625 (0.010)** |
| **aayl49_ML** | 0.518 (0.008) | 0.524 (0.008) | 0.487 (0.009) | 0.320 (0.009) | 0.499 (0.007) | 0.524 (0.007) | **0.531 (0.007)** |
| **aayl51** | 0.576 (0.007) | 0.516 (0.009) | 0.524 (0.008) | 0.260 (0.011) | 0.527 (0.009) | 0.566 (0.010) | **0.579 (0.011)** |

## 4.4 CONTROLLABLE SAMPLING OF ANTIBODY LOOPS

In this section, we evaluate IGLOOALM on its ability to guide the structure of the loop at the residue level, by analyzing if sampled loops are consistent in structure to the masked out loop.

**Experimental setup.** For the CDR1, CDR2, and CDR3 of the heavy and light chain, we randomly sample 50 structures from SAbDab, which are in the test set of IGLOO. IGLOO token, $\mathbf{t}$, and the dihedral angles and masked sequence $\mathbf{X}$ are provided as input to IGLOOALM. We sample loop sequences from the resulting amino acid likelihoods of IGLOOALM for each of these antibodies.

**Baselines.** We evaluate against recently published state-of-the-art inverse folding models for antibodies: AbMPNN (Dreyer et al., 2023)–a version of ProteinMPNN (Dauparas et al., 2022) finetuned on antibody structures, and AntiFold (Høie et al., 2024)–a version of ESM-IF1 (Hsu et al., 2022) finetuned on antibody structures. For these models, we only generate the loop sequence given the full backbone and the sequence of the rest of the antibody. We also compare to the base model, IgBert, and IGLOOLM which does not include the multimodal residue tokens $\mathbf{X}$.

**Evaluation setup.** For each loop and model, we sample 10 sequences at the following sampling temperatures: $\lambda = 0.01, 0.05, 0.1, 0.2, 0.5, 1.0, 2.0$. In total, for each model, we generate $50 \times 10 \times 6 \times 7 = 21,000$ sequences for the different structures, sequence samples, loop types, and temperatures, respectively. Then we align the generated loop regions with the original structure and evaluate the self-consistency (sc) RMSD between the two loop structures. Sampling sequences at different temperatures is necessary to generate sequences with different levels of sequence identity to the original loop, since recapitulating the original loop sequence would achieve low scRMSD but would not be useful for the design of new H3 loops.

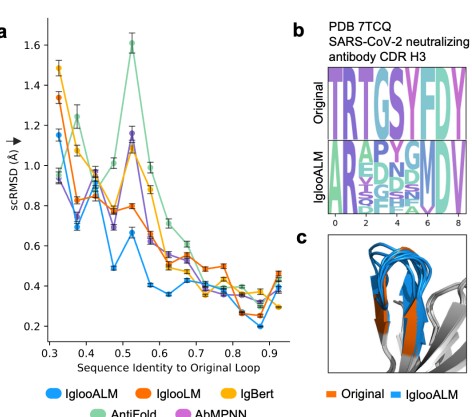

Figure 4: **a** Mean self-consistency (sc) RMSD (Å) ↓ of sampled loop sequences compared to original loop structures across sequence identity bins. Error bars show standard error of the mean across the generated structures aggregated in each sequence identity bin. **b** Sequence logo of original and ten IGLOOALM sampled sequences of the CDR H3 loop region for a SARS-CoV-2 neutralizing antibody (PDB 7TCQ) at $\lambda = 0.5$. **c** Predicted structure of the CDR H3 loop regions aligned to PDB 7TCQ.

**Results.** We show the scRMSD of the sampled loops stratified by sequence identity (Fig. 4a). IGLOOALM excels at generating loops at different levels of sequence diversity while maintaining a similar structure, improving both on state-of-the-art antibody inverse folding models and the base model. Notably, Igloo outperforms AbMPNN for 9 out of 13 sequence-identity bins. We show the redesign of the H3 loop of a SARS-CoV-2 neutralizing antibody from the PDB structure 7TCQ.

At a sampling temperature of 0.5, IGLOOALM samples loops with an average edit distance of 6.6 from the loop of length 9 (Fig. 4b). The predicted structures of the sampled loops maintain the beta hairpin structure of the original loop with an average loop RMSD of 0.79 Å (Fig. 4c). Additional examples of sampled loops and their predicted structures are in Fig. S3.

### 4.5 PRIORITIZATION OF BINDERS FROM HIGH-THROUGHPUT LIBRARIES WITH IGLOO

We demonstrate an application of IGLOO for prioritizing CDR sequences in hit-to-lead optimization. Given a hit antibody, we denote its sequence as the *seed sequence*. Exploring variants of CDR loops is a key step in diversifying the hit library. However, because such libraries can contain up to $10^{10}$ variants (Ponsel et al., 2011), exhaustive evaluation with large foundation models is computationally infeasible. For a given diversified library, we show that IGLOO can efficiently and scalably identify potential binders by nominating sequences that share the same quantized IGLOO token, $\hat{t}$, as the seed sequence.

**Experimental Setup.** We evaluate IGLOO using the CDR H3 loop library of length 15 targeting HER2, introduced in Mason et al. (2021). This library was generated through combinatorial mutagenesis and experimentally screened using fluorescence-activated cell sorting and deep sequencing, yielding 38,860 unique CDR H3 loops comprising 11,309 binders and 27,551 non-binders. Our objective is to test whether IGLOO can be used in a *zero-shot* setting to prioritize experimentally validated binders. We apply IGLOO to all 38,860 loops to obtain the quantized token $\hat{t}$ for each sequence. Binders are prioritized by selecting loops whose $\hat{t}$ matches that of the seed sequence.

**Results.** Among loops with IGLOO tokens determined by sequence alone match the seed, 55.3% are experimentally validated binders, representing a $1.9\times$ enrichment compared to the baseline proportion of 29.1% in the full library. When incorporating sequence and structure-based tokens for loops predicted with Ibex, the proportion of binders among matching loops is similarly high at 47.4%. Figure 5 further stratifies binder enrichment by edit distance to the seed, showing that IGLOO consistently achieves higher precision for loops at different thresholds of sequence identity to the seed. In practice, IGLOO-based screening is highly scalable. Processing the entire library of 38,860 loops required only 10 minutes using sequence-based tokens.

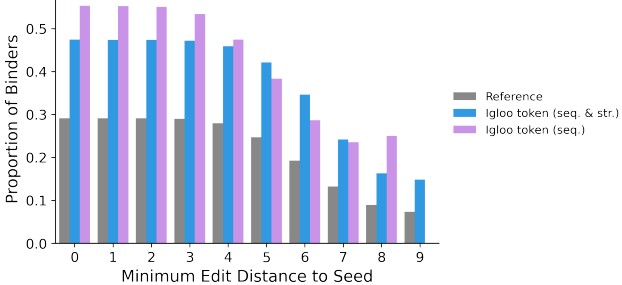

Figure 5: Proportion of binders stratified by a minimum edit distance cutoff nominated by IGLOO when selecting for CDR H3 loops with the same quantized token, $\hat{t}$, as the CDR H3 loop of the seed binder for HER2. The reference is the proportion of binders in the library.

## 5 CONCLUSION

Here we present IGLOO, a multimodal tokenizer for antibody loops with a novel contrastive learning objective based on dihedral angle distance between loop backbones. Applying IGLOO, we achieve state-of-the-art results in retrieving similar loop conformations and recover known canonical clusters. IGLOO tokens can also be incorporated into protein language models for improved binding affinity predictions with IGLOOLM and for controllable generation of antibody loops with IGLOOALM. While IGLOOALM demonstrates strong in silico results, more comprehensive wet-lab validation evaluation is needed to evaluate whether redesigned antibodies maintain binding with antigens. IGLOO could be further extended to incorporate other modalities such as all-atom structure, epitope information, function, and binding affinity. By introducing multimodal loop tokens, IGLOO opens new directions for multimodal foundation models for rational antibody design.

REPRODUCIBILITY STATEMENT

All code for data processing, implementation, training scripts, evaluation and analyses scripts, and for reproducing results in the paper is available at the anonymized repository `https://anonymous.4open.science/r/igloo`. Details of the dataset used for training IGLOO are provided in Appendix A.1, and for training IGLOOLM and IGLOOALM are provided in Appendix A.2. Details for processing of AbBiBench data are available at Appendix A.3. Training details, hyperparameters, GPUs used, and training duration for training IGLOO, IGLOOLM, IGLOOALM, and regression models for AbBiBench are available at Appendix B. The transformer architecture used in IGLOO is based on the `TransformerLayer` from ESM-2 available at `https://github.com/facebookresearch/esm`. The base model, IgBert (Kenlay et al., 2024), is publicly available at `https://huggingface.co/Exscientia/IgBert`. Code for finetuning IgBert for IGLOOLM and IGLOOALM is available on our GitHub. Full details of the IGLOO loss function and objective are available at Section 3.2. All evaluation metrics used for experiments are specified explicitly in Section 4.

ETHICS STATEMENT

IGLOO is a method for tokenizing loop regions of antibodies and may be used for antibody design. All antibody sequences used to develop and evaluate IGLOO were obtained from publicly available databases and contain no personal or patient-identifiable information. No new animal or human subjects were involved. Methods that facilitate antibody engineering can present dual-use concerns. Here we present use cases where IGLOO is applied for achieving positive impact.

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

## LLM USAGE

ChatGPT and Gemini were used to aid and polish writing. GitHub Copilot was used to aid code autocompletion for producing results outlined in the paper.

## A  DATASET PROCESSING

### A.1  IGLOO TRAINING DATA

We process 18,303 structures from SAbDab and STCRDab, which are comprised of 14,341 antibodies, 3,095 nanobodies, and 867 TCRs. From these structures, we run ANARCI (Dunbar & Deane, 2016) on the sequences to identify the loop regions (CDR1, CDR2, CDR3, CDR4) in the North definition North et al. (2011) from their AHo alignment (Honegger & Pluèckthun, 2001). A valid loop requires defined $\phi, \psi, \omega$ angles and at least 5 residues before and after the loop, referred to as the stem region, yielding 108,167 loop structures. To define a train, test, and validation split, we cluster on the concatenated CDR sequences with MMseqs2 (Steinegger & Söding, 2017), using an 80% sequence identity threshold.

Ibex (Dreyer et al., 2025) predicted CDR loops from antibodies in paired OAS are also included in the IGLOO training set. To maximize sequence diversity of the predicted loop structures, they are downsampled from an initial set of 2,447,258 down to 87,456 by clustering on both concatenated CDR sequences as well as H3 loop sequences with MMseqs2 (Steinegger & Söding, 2017), using a 50% sequence identity threshold. In total, we include 699,648 predicted loop structures in the training set.

Table S1: Number of each loop type in the IGLOO training dataset from SAbDab, STCRDab and paired OAS.

| Loop Type | SAbDab and STCRDab | Paired OAS |
|---|---|---|
| H1 | 14,877 | 87,456 |
| H2 | 14,876 | 87,456 |
| H3 | 14,875 | 87,456 |
| H4 | 14,877 | 87,456 |
| L1 | 12,167 | 87,456 |
| L2 | 12,168 | 87,456 |
| L3 | 12,159 | 87,456 |
| L4 | 12,168 | 87,456 |
| Total | 108,167 | 699,648 |

### A.2  IGLOOLM AND IGLOOALM TRAINING DATA

We fold the heavy and light chains with Ibex for 2,447,258 antibodies. The heavy chains and light chains are clustered separately with MMseqs2 (Steinegger & Söding, 2017), using a 90% sequence

identity threshold. This results in 247,156 light chain clusters and 875,767 heavy chain clusters. We randomly sample 10,000 light chain and 20,000 heavy chain clusters for the validation and test sets, respectively. For the training set, we keep all sequences in the sequence identity clusters, and for the validation and test sets, we only keep the representative sequence from each cluster. In total, we train IGLOOLM and IGLOOALM on 4,598,332 antibody chains.

## A.3 AʙBɪBᴇɴᴄʜ Data

We use AbBiBench (Zhao et al., 2025) benchmark, which has for an antibody-antigen pair, heavy chain mutant sequences and their binding affinity score. The binding affinity score is the $-\log K_d$ for all antibody-antigen pairs except for 2fjg and 1mlc, which is $\log$ enrichment. For some antibody-antigens, we filter out sequences that do not have binding affinity scores and are given default scores instead. The final number of sequences and filtered used for each antibody-antigen target is shown in Table S2. We train models for 10 out of 11 antibody-antigens in AbBiBench. The Integrin-$\alpha$-1 AQC2 antibody-antigen dataset is not tested due to an insufficient number of binding affinity measurements ($N = 40$). For each sequence, we fold the heavy chain with the light chain from the structure in the PDB ID with Ibex and extract the structures of the loops for IGLOOLM. All binding affinity values for an antibody-antigen pair are scaled by subtracting the mean of the training distribution and scaling to unit variance.

Table S2: Number of sequences for each antibody-antigen in the AbBiBench benchmarking dataset.

| PDB ID | Seed antibody | Antigen | Number of sequences | Filtered out values |
|---|---|---|---|---|
| 1n8z | Trastuzumab | HER2 | 419 | - |
| 1mlc | D44.1 | Hen-egg-white lysozyme | 1,229 | - |
| 2fjg | G6.31 | VEGF | 2,223 | - |
| 3gbn_h1 | CR6261 | Influenza A/New Caledonia/20/99 (H1N1) | 1,673 | 7.0 |
| 3gbn_h9 | CR6261 | Influenza A/Hong Kong/1073/1999 (H9N2) | 1,470 | 7.0 |
| 4fqi_h1 | CR9114 | Influenza A/New Caledonia/20/99 (H1N1) | 63,419 | 7.0 |
| 4fqi_h3 | CR9114 | Influenza A/Wisconsin/67/2005 (H3N2) | 7,174 | 6.0 |
| aayl49 | AAYL49 | Spike HR2 | 4,312 | - |
| aayl49_ML | AAYL49_ML | Spike HR2 | 8,953 | - |
| aayl51 | AAYL51 | Spike HR2 | 4,320 | - |

## B IMPLEMENTATION DETAILS

### B.1 TRAINING IGLOO

IGLOO is trained for 100 epochs on 1 NVIDIA H100. We set the following hyperparameters for training IGLOO to be dihedral temperature (0.1), unit circle regularization weight (0.01), number of transformer layers (4), codebook commit loss weight (0.5), max loop length (36), and batch size (64). The following hyperparameters were chosen from: learning rate ($10^{-5} - 10^{-3}$), embedding dimension (32, 128, 1024), codebook size (1024, 8192), and weight decay ($0, 10^{-5}$). We select the checkpoint at the epoch with the lowest validation loss and select the best hyperparameter based on the average recovery of the canonical clusters (Kelow et al., 2022) on the validation set. We use a two-phase training approach; in the first phase, the model is trained on the SAbDab and paired OAS dataset, and in the second phase, the model is trained only on SAbDab. We use an embedding dimension of 128 and a codebook size of 8192. In the first stage, the learning rate is $5 \times 10^{-5}$ and weight decay 0, and for the second stage, a learning rate of $5 \times 10^{-5}$ and weight decay of $10^{-5}$ is used.

### B.2 TRAINING IGLOOLM AND IGLOOALM

We take the publicly available pretrained weights and hyperparameters from IgBert (Kenlay et al., 2024) and continue to finetune the model with the IGLOO tokens for IGLOOLM, and with IGLOO tokens and multimodal residue tokens for IGLOOALM. Both models were trained for 3 days on 4 NVIDIA H100s, which correspond to 53k steps over 5 epochs.

### B.3 PARATOPE RETRIEVAL

For obtaining embeddings from the baseline models ESM C, ESM-2 (3B), SaProt, ProstT5, AbLang2, and IgBert, we use publicly available weights and embed the whole antibody chain and use the mean embedding of the loop residues as the loop embedding. For the structure tokenizers MIF, ProteinMPNN, and Amino Aseed (continuous tokens) we use the implementation provided by (Yuan et al., 2025) and average residue-level tokens over the loop region to obtain a loop embedding. For Foldseek 3Di, since the embeddings are only 2-dimensional, we concatenate the flattened representations of all of the residues in the loop region to obtain the loop embedding.

### B.4 TRAINING REGRESSION MODELS FOR ABBIBENCH

For the embeddings of each model and antibody-antigen target, we train a ridge regression with a 10-fold nested cross-validation. The 10 outer folds are used for testing, each containing a 5-fold inner cross-validation that selects the optimal L2 penalty $\lambda \in \{1, 10^{-1}, 10^{-2}, \ldots, 10^{-6}, 0\}$ within that fold. For every outer fold, the model was retrained with its fold-specific best $\lambda$ on the entire training partition, scored on the held-out test partition, and the Spearman correlation coefficient, $\rho$, is averaged across the 10 folds.

## C ABLATION STUDY

To evaluate the contributions of the components of IGLOO, we conduct the following ablation studies to understand the effect of (1) the dihedral distance contrastive loss, (2) distance-threshold filtering of positive ($\mathcal{D} < 0.1$) and negative ($\mathcal{D} > 0.47$) pairs, (3) the sequence modality track, (4) the dihedral angle modality track, and (5) only defining positive pairs between loops of the same length (Appendix D). We evaluate the ablated models with the same experimental setup as outlined in Section 4.1.

Table S3: Average precision at rank 1, 5, 10, and 20 for retrieval of similar CDR paratopes evaluated with RMSD $< 1$Å and $\mathcal{D} < 0.47$. The **first** and second best performance are highlighted below. CL is contrastive learning, DT filter is distance-threshold filter of positive ($\mathcal{D} < 0.1$) and negative ($\mathcal{D} > 0.47$) pairs, and loop length refers to training IGLOO with positive pairs defined between loops of different lengths (Appendix D).

| # Loops retrieved | | % RMSD $< 1$ Å | | | | | | % $\mathcal{D} < 0.47$ | | | | | |
| | | L1 | L2 | L3 | H1 | H2 | H3 | L1 | L2 | L3 | H1 | H2 | H3 |
|---|---|---|---|---|---|---|---|---|---|---|---|---|---|
| 1 | Random | 0.518 | 0.603 | 0.417 | 0.221 | 0.321 | 0.152 | 0.669 | 0.770 | 0.459 | 0.597 | 0.435 | 0.157 |
| | No CL loss | 0.866 | 0.732 | *0.714* | 0.418 | 0.697 | 0.305 | 0.915 | *0.967* | *0.836* | 0.686 | *0.898* | 0.464 |
| | No DT filter | 0.877 | **0.748** | 0.673 | 0.452 | 0.680 | 0.330 | 0.919 | **0.994** | 0.789 | 0.862 | 0.882 | 0.539 |
| | Sequence only | 0.790 | 0.740 | 0.666 | 0.413 | 0.472 | 0.219 | 0.809 | 0.956 | 0.719 | 0.750 | 0.557 | 0.292 |
| | Dihedral angles only | 0.870 | 0.702 | 0.591 | *0.557* | 0.590 | 0.298 | **0.945** | 0.936 | 0.752 | *0.841* | 0.741 | 0.491 |
| | Mismatched length | **0.878** | *0.734* | 0.749 | 0.593 | **0.701** | **0.339** | 0.941 | 0.993 | **0.870** | 0.834 | **0.928** | *0.523* |
| | IGLOO | *0.871* | **0.748** | **0.761** | **0.603** | *0.691* | *0.327* | *0.935* | 0.993 | 0.856 | **0.885** | 0.918 | **0.669** |
| 5 | Random | 0.558 | 0.556 | 0.387 | 0.229 | 0.347 | 0.138 | 0.679 | 0.742 | 0.401 | 0.561 | 0.480 | 0.136 |
| | No CL loss | 0.826 | 0.724 | *0.636* | 0.420 | 0.611 | 0.279 | *0.897* | *0.973* | 0.765 | 0.677 | 0.881 | 0.413 |
| | No DT filter | *0.829* | **0.748** | 0.629 | 0.453 | 0.646 | 0.315 | 0.904 | **0.996** | *0.784* | 0.788 | 0.876 | 0.506 |
| | Sequence only | 0.798 | 0.704 | 0.569 | 0.401 | 0.477 | 0.203 | 0.844 | 0.909 | 0.663 | 0.633 | 0.573 | 0.245 |
| | Dihedral angles only | 0.837 | 0.682 | 0.611 | *0.463* | 0.565 | *0.280* | 0.894 | 0.917 | 0.759 | *0.786* | 0.728 | 0.454 |
| | Mismatched length | 0.828 | *0.740* | 0.646 | 0.473 | *0.627* | **0.316** | *0.897* | 0.993 | **0.828** | 0.643 | 0.833 | *0.495* |
| | IGLOO | **0.841** | 0.743 | **0.666** | **0.501** | **0.658** | 0.315 | **0.909** | 0.993 | 0.827 | **0.805** | **0.923** | **0.553** |
| 10 | Random | 0.550 | 0.556 | 0.384 | 0.235 | 0.345 | 0.132 | 0.666 | 0.726 | 0.399 | 0.556 | 0.499 | 0.133 |
| | No CL loss | 0.796 | 0.723 | *0.601* | 0.424 | 0.559 | 0.265 | 0.866 | 0.975 | *0.724* | 0.672 | *0.795* | 0.376 |
| | No DT filter | 0.802 | **0.744** | 0.584 | *0.460* | 0.600 | 0.302 | **0.884** | **0.994** | 0.717 | *0.728* | 0.840 | 0.470 |
| | Sequence only | 0.781 | 0.705 | 0.503 | 0.391 | 0.494 | 0.198 | 0.852 | 0.898 | 0.571 | 0.655 | 0.613 | 0.239 |
| | Dihedral angles only | **0.812** | 0.672 | 0.595 | 0.442 | 0.541 | 0.263 | 0.880 | 0.905 | 0.715 | **0.757** | 0.710 | 0.400 |
| | Mismatched length | *0.804* | *0.737* | 0.604 | 0.469 | *0.564* | **0.304** | 0.867 | 0.992 | 0.735 | 0.671 | 0.759 | *0.462* |
| | IGLOO | 0.809 | 0.742 | **0.623** | **0.473** | **0.620** | *0.300* | 0.879 | 0.993 | **0.764** | 0.736 | **0.854** | **0.473** |
| 20 | Random | 0.545 | 0.557 | 0.373 | 0.249 | 0.351 | 0.127 | 0.648 | 0.730 | 0.392 | 0.559 | 0.508 | 0.126 |
| | No CL loss | 0.780 | 0.689 | 0.516 | 0.410 | 0.496 | 0.242 | 0.845 | 0.927 | 0.603 | 0.649 | 0.700 | 0.335 |
| | No DT filter | 0.788 | 0.704 | 0.543 | 0.459 | 0.562 | 0.279 | **0.860** | *0.950* | 0.635 | 0.686 | *0.747* | **0.417** |
| | Sequence only | 0.761 | 0.693 | 0.484 | 0.400 | 0.482 | 0.193 | 0.828 | 0.887 | 0.533 | 0.614 | 0.624 | 0.217 |
| | Dihedral angles only | **0.795** | 0.651 | 0.524 | 0.424 | 0.497 | 0.245 | 0.853 | 0.884 | 0.617 | 0.702 | 0.671 | 0.356 |
| | Mismatched length | *0.789* | *0.703* | *0.533* | **0.465** | *0.538* | **0.280** | 0.843 | 0.954 | 0.627 | 0.689 | 0.749 | *0.408* |
| | IGLOO | 0.793 | **0.705** | **0.558** | 0.459 | **0.578** | 0.278 | *0.851* | **0.956** | **0.674** | **0.715** | **0.749** | *0.402* |

**Dihedral distance contrastive loss.** Key to the IGLOO approach is the contrastive learning objective for the model to learn to place loops that share similar backbone dihedral angles in the same region of the latent space. To test this component of the model, we removed the dihedral contrastive loss from training. Consequently, the ablated model is only focused on the reconstruction of masked amino acids and dihedral angles. We observed in Table S3 that the contrastive learning objective improves performance across loop regions on both precision for RMSD $< 1\text{Å}$ and $\mathcal{D} < 0.47$, with improvements of 11.8% on the L3 loop and 20.0% on the H3 loop in precision at rank 20.

**Distance-threshold filtering of positive ($\mathcal{D} < 0.1$) and negative ($\mathcal{D} > 0.47$) pairs.** $\mathcal{D}$ is a continuous measure of the difference in dihedral angles between two loop backbones. In order for the model to not overfit to an arbitrary threshold of 0.47, which was established by Kelow et al. (2022), we established a distance-threshold filter where loops with $0.1 \leq \mathcal{D} \leq 0.47$ are ignored. We train an ablated model where positive ($\mathcal{D} \leq 0.47$) and negative ($\mathcal{D} > 0.47$) pairs and find that performance is generally comparable to when distance-thresholding is applied, with IGLOO offering slight improvements across most loop types.

**Multimodal learning in IGLOO.** In IGLOO the input to the transformer is $\mathbf{X} = \mathbf{D} + \mathbf{A}$, in this section we remove the dihedral angles, $\mathbf{D}$, and sequence, $\mathbf{A}$, from the model separately. We also adjust the loss function correspondingly. The masked reconstruction of dihedral angles and the masked reconstruction of amino acid identities objectives are also removed, respectively. In Table S3, we observe the dihedral angle modality is most important to the retrieval task, notably for the H3 loop retrieval with an improvement of 85.2%. The addition of the sequence modality is also helpful with improvements observed for almost all loop types and on H3 loop retrieval, an improvement of 12.7% is observed.

## D   IGLOO WITH MISMATCHED LOOP LENGTH

The IGLOO contrastive loss function only assigns positive labels to pairs of loops of the same length. However, Nowak et al. (2016) explore CDR clusters with loops of multiple lengths, and find clusters L1–10,11,12-A; L1–13,14-A; L3–9,10-A; and L3–10,11-A with loops of different lengths. In this section, we show how IGLOO can be trained to align loops of different lengths in the latent space.

To define positive pairs for loops of different lengths, we use the approach from Nowak et al. (2016). Loops are aligned by the $C_\alpha$ coordinates of their stem region, which we define as the $N_{\text{stem}}$ amino acids before and after the loop. We then use dynamic time warping (DTW) (Giorgino, 2009) to determine an alignment between the loop $C_\alpha$ coordinates. For the aligned residues, the dihedral distance $\mathcal{D}$ is calculated (Algorithm 1). Finally, positive and negative pairs are defined by thresholds on $\mathcal{D}$.

We train the IGLOO architecture with $N_{\text{stem}} = 5$ and tolerance $k = 1$, which is consistent with the multi-length clusters found by Nowak et al. (2016). For batches in an epoch, we find on average 5.0% of pairs of loops in the batch to be of different lengths and $\mathcal{D} < 0.1$. In Table S3 we observe that training IGLOO with positive pairs defined between loops of different lengths leads to a slight decay in performance on most loop types.

---

**Algorithm 1** Dihedral angle distance between a pair of loops of different lengths

---

**Input:** loop dihedral angles $\phi_1, \psi_1, \omega_1 \in (-\pi, \pi]^n$, $\phi_2, \psi_2, \omega_2 \in (-\pi, \pi]^m$; loop $C_\alpha$ coordinates $\mathbf{L}_1 \in \mathbb{R}^{n \times 3}$, $\mathbf{L}_2 \in \mathbb{R}^{m \times 3}$; stem $C_\alpha$ coordinates $\mathbf{S}_1, \mathbf{S}_2 \in \mathbb{R}^{N_{\text{stem}} \times 3}$; tolerance $k$ (max. residue length difference)

**Output:** dihedral angle distance $\mathcal{D} \in [0, 4]$

1: **if** $|n - m| > k$ **then**
2:      **return** `4.0`                                                 $\triangleright$ returns $\mathcal{D}_{\max}$
3: **end if**
4: $\boldsymbol{\mu}_1 \leftarrow \text{mean}(\mathbf{S}_1);$     $\boldsymbol{\mu}_2 \leftarrow \text{mean}(\mathbf{S}_2)$
5: $\tilde{\mathbf{S}}_1 \leftarrow \mathbf{S}_1 - \boldsymbol{\mu}_1;$     $\tilde{\mathbf{S}}_2 \leftarrow \mathbf{S}_2 - \boldsymbol{\mu}_2$
6: $(\mathbf{R}, \boldsymbol{\tau}, \text{RMSD}_{\text{stem}}) \leftarrow \text{Kabsch}(\tilde{\mathbf{S}}_1, \tilde{\mathbf{S}}_2)$
7: **if** $\text{RMSD}_{\text{stem}} > 1.0$ Å **then**
8:      **return** `4.0`                                                 $\triangleright$ returns $\mathcal{D}_{\max}$
9: **end if**
10: $\tilde{\mathbf{L}}_1 \leftarrow (\mathbf{L}_1 - \boldsymbol{\mu}_1)\mathbf{R}^\top + \boldsymbol{\mu}_2 + \boldsymbol{\tau};$     $\tilde{\mathbf{L}}_2 \leftarrow \mathbf{L}_2$
11: $\mathcal{P} \leftarrow \text{DTW}(\tilde{\mathbf{L}}_1, \tilde{\mathbf{L}}_2)$                 $\triangleright$ warping path $\mathcal{P}$ mapping residues of $\mathbf{L}_1$ to $\mathbf{L}_2$
12: $\tilde{\phi}_1 \leftarrow \mathcal{P}(\phi_1);$     $\tilde{\psi}_1 \leftarrow \mathcal{P}(\psi_1);$     $\tilde{\omega}_1 \leftarrow \mathcal{P}(\omega_1);$
13: $\mathcal{D}_\phi \leftarrow \text{mean}\big(2\big(1 - \cos(\tilde{\phi}_1 - \phi_2)\big)\big);$     $\mathcal{D}_\psi \leftarrow \text{mean}\big(2\big(1 - \cos(\tilde{\psi}_1 - \psi_2)\big)\big);$     $\mathcal{D}_\omega \leftarrow$ $\text{mean}\big(2\big(1 - \cos(\tilde{\omega}_1 - \omega_2)\big)\big)$
14: $\mathcal{D} \leftarrow \text{mean}(\mathcal{D}_\phi, \mathcal{D}_\psi, \mathcal{D}_\omega)$
15: **return** $\mathcal{D}$

---

# E ADDITIONAL RESULTS

## E.1 LOOP RECONSTRUCTION

Table S4: Amino acid recovery and dihedral angle prediction performance of IGLOO across SAbDab test set loops at a masking rate of 30%. Dihedral angle prediction performance is evaluated with 4 (MACE).

| Loop | AA Recovery (%) | MACE (°) | | |
|---|---|---|---|---|
| | | $\phi$ | $\psi$ | $\omega$ |
| L1 | 69.41 | 14.29 | 14.28 | 3.89 |
| L2 | 60.13 | 10.72 | 13.64 | 3.71 |
| L3 | 54.19 | 18.55 | 20.50 | 5.72 |
| L4 | 60.95 | 11.42 | 11.77 | 3.54 |
| H1 | 63.82 | 16.83 | 20.03 | 4.25 |
| H2 | 53.42 | 14.24 | 14.59 | 4.08 |
| H3 | 41.67 | 27.09 | 33.48 | 5.05 |
| H4 | 55.63 | 11.16 | 11.99 | 3.91 |

## E.2 PARATOPE RETRIEVAL

We present additional results for the retrieval of similar loops evaluated with precision at rank 1, 5, and 10 (Table S5).

Table S5: Average precision at rank 1, 5, and 10 for retrieval of similar CDR paratopes. Models are shown in rows. The **first**, second, and *third* best performance for each column are highlighted.

| # Loops retrieved | | % RMSD < 1 Å | | | | | | % $\mathcal{D}$ < 0.47 | | | | | |
|---|---|---|---|---|---|---|---|---|---|---|---|---|---|
| | | L1 | L2 | L3 | H1 | H2 | H3 | L1 | L2 | L3 | H1 | H2 | H3 |
| | ESM C | 0.765 | 0.747 | 0.561 | 0.420 | 0.670 | 0.242 | 0.860 | 0.969 | 0.533 | 0.549 | 0.786 | 0.254 |
| | ESM-2 (3B) | 0.706 | 0.733 | 0.525 | 0.420 | 0.586 | 0.238 | 0.793 | 0.960 | 0.505 | 0.566 | 0.867 | 0.228 |
| | AbLang2 | 0.766 | 0.747 | 0.535 | 0.384 | 0.567 | 0.191 | 0.835 | 0.919 | 0.533 | 0.575 | 0.825 | 0.212 |
| | IgBert | 0.802 | 0.733 | 0.621 | 0.410 | 0.428 | 0.194 | 0.853 | 0.971 | 0.597 | 0.528 | 0.653 | 0.252 |
| | SaProt | 0.804 | 0.747 | 0.696 | 0.475 | 0.525 | 0.271 | 0.877 | 0.969 | 0.722 | 0.733 | 0.674 | 0.341 |
| 1 | ProstT5 | 0.791 | **0.748** | 0.729 | *0.531* | *0.703* | 0.331 | *0.920* | 0.987 | **0.878** | 0.748 | 0.871 | 0.487 |
| | MIF | 0.879 | 0.747 | 0.643 | 0.557 | 0.564 | 0.290 | 0.915 | 0.991 | 0.806 | 0.911 | 0.837 | 0.381 |
| | ProteinMPNN | 0.863 | 0.733 | 0.706 | 0.468 | 0.693 | *0.355* | 0.906 | 0.958 | 0.798 | **0.915** | *0.900* | *0.543* |
| | Foldseek 3Di | 0.840 | 0.748 | 0.739 | 0.509 | **0.713** | 0.361 | **0.947** | **0.994** | 0.859 | 0.771 | 0.874 | 0.585 |
| | Amino Aseed | **0.890** | 0.734 | *0.738* | 0.462 | *0.703* | **0.374** | 0.907 | **0.994** | 0.824 | 0.787 | *0.910* | 0.528 |
| | IGLOO | *0.871* | 0.748 | **0.761** | **0.603** | 0.691 | 0.327 | 0.935 | 0.993 | *0.856* | *0.885* | **0.918** | **0.669** |
| | ESM C | 0.765 | 0.710 | 0.610 | 0.408 | 0.627 | 0.214 | 0.842 | 0.900 | 0.619 | 0.658 | 0.748 | 0.256 |
| | ESM-2 (3B) | 0.730 | 0.711 | 0.540 | 0.447 | 0.562 | 0.225 | 0.837 | 0.910 | 0.607 | 0.697 | 0.788 | 0.251 |
| | AbLang2 | 0.720 | 0.717 | 0.588 | 0.402 | 0.563 | 0.188 | 0.786 | 0.896 | 0.595 | 0.604 | 0.816 | 0.238 |
| | IgBert | 0.776 | 0.709 | 0.570 | 0.397 | 0.578 | 0.182 | 0.845 | 0.938 | 0.566 | 0.677 | 0.798 | 0.259 |
| | SaProt | 0.782 | 0.741 | 0.607 | 0.441 | 0.552 | 0.246 | 0.858 | 0.970 | 0.654 | 0.698 | 0.753 | 0.316 |
| 5 | ProstT5 | 0.805 | **0.745** | 0.643 | *0.499* | **0.671** | 0.302 | 0.879 | *0.985* | 0.772 | 0.742 | *0.841* | 0.445 |
| | MIF | 0.825 | 0.739 | 0.652 | 0.470 | 0.571 | 0.263 | 0.886 | 0.985 | 0.787 | 0.755 | 0.750 | 0.361 |
| | ProteinMPNN | 0.851 | 0.742 | 0.651 | 0.482 | 0.648 | *0.327* | 0.878 | 0.974 | 0.787 | *0.762* | 0.836 | *0.472* |
| | Foldseek 3Di | 0.810 | 0.742 | **0.666** | **0.530** | **0.680** | 0.326 | 0.886 | 0.985 | 0.815 | 0.784 | 0.869 | 0.475 |
| | Amino Aseed | **0.852** | 0.745 | 0.657 | 0.469 | 0.606 | **0.336** | 0.870 | **0.994** | *0.813* | 0.736 | 0.825 | 0.469 |
| | IGLOO | *0.841* | *0.743* | **0.666** | 0.501 | *0.658* | 0.315 | **0.909** | 0.993 | **0.827** | **0.805** | **0.923** | **0.553** |
| | ESM C | 0.768 | 0.705 | 0.537 | 0.418 | 0.583 | 0.202 | 0.828 | 0.921 | 0.572 | 0.678 | 0.768 | 0.230 |
| | ESM-2 (3B) | 0.744 | 0.713 | 0.521 | 0.439 | 0.561 | 0.210 | 0.837 | 0.895 | 0.569 | 0.716 | 0.734 | 0.237 |
| | AbLang2 | 0.719 | 0.686 | 0.522 | 0.408 | 0.553 | 0.186 | 0.799 | 0.844 | 0.558 | 0.611 | 0.764 | 0.228 |
| | IgBert | 0.727 | 0.691 | 0.520 | 0.416 | 0.542 | 0.182 | 0.806 | 0.920 | 0.541 | 0.708 | 0.726 | 0.245 |
| | SaProt | 0.749 | 0.733 | 0.561 | 0.439 | 0.551 | 0.237 | 0.816 | 0.961 | 0.653 | 0.722 | 0.730 | 0.277 |
| 10 | ProstT5 | 0.799 | **0.744** | 0.597 | 0.499 | *0.615* | 0.291 | *0.870* | 0.974 | 0.713 | 0.731 | 0.771 | 0.404 |
| | MIF | 0.796 | 0.732 | 0.606 | 0.463 | 0.529 | 0.253 | 0.864 | 0.984 | 0.719 | 0.714 | 0.675 | 0.337 |
| | ProteinMPNN | 0.821 | 0.742 | 0.601 | *0.481* | 0.570 | *0.310* | 0.865 | 0.980 | 0.719 | **0.751** | 0.765 | *0.419* |
| | Foldseek 3Di | 0.800 | 0.728 | **0.630** | **0.508** | **0.635** | *0.307* | 0.873 | 0.959 | 0.750 | **0.751** | *0.797* | 0.416 |
| | Amino Aseed | **0.830** | 0.742 | *0.619* | 0.453 | 0.599 | **0.318** | 0.869 | **0.993** | *0.738* | 0.717 | 0.822 | 0.427 |
| | IGLOO | *0.809* | 0.742 | 0.623 | 0.473 | 0.620 | 0.300 | **0.879** | **0.993** | **0.764** | 0.736 | **0.854** | **0.473** |

### E.3 LOOPS WITH NO KNOWN CANONICAL CLUSTER

Table S6: Proportion of loops in SAbDab with no known Kelow et al. (2022) canonical cluster with a cutoff of $\mathcal{D} = 0.47$ to cluster centroids.

| CDR | Heavy | Light |
|---|---|---|
| CDR1 | 0.130 | 0.112 |
| CDR2 | 0.098 | 0.187 |
| CDR3 | 0.763 | 0.192 |
| CDR4 | 0.037 | 0.062 |

### E.4 VISUALIZATION OF IGLOO LATENT SPACE

In Fig. S1 we visualize the IGLOO token $\mathbf{t}$ for all loops in SAbDab across train, test, and validation datasets in 2D with Uniform Manifold Approximation and Projection (UMAP) (McInnes et al., 2018). We observe in the UMAP that the embeddings are localized by their loop type, loop length, and canonical cluster. Among the CDRs, the H3 embeddings span the broadest region of the UMAP manifold, reflecting their markedly higher structural diversity.

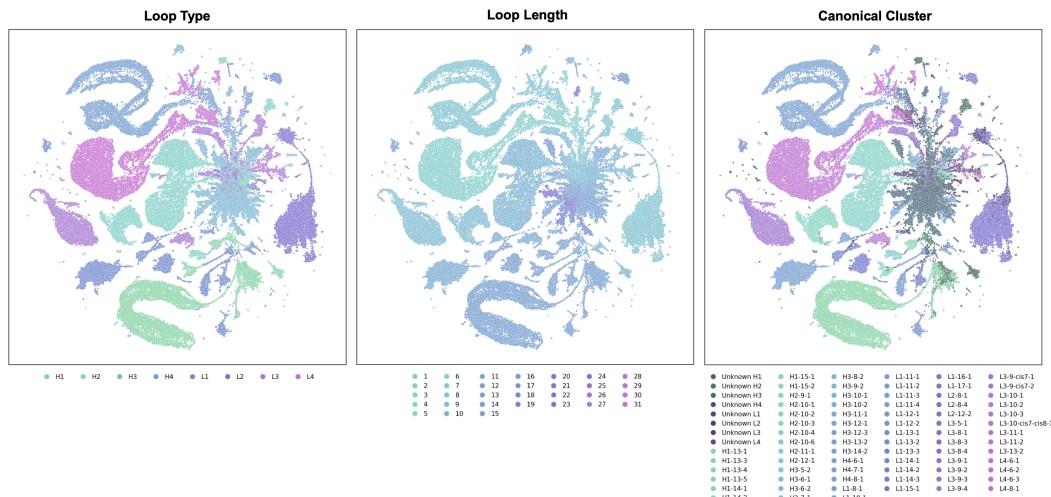

Figure S1: UMAP of the IGLOO latent space for loops in SAbDab. **Left** Loops colored by loop type. **Middle** Loops colored by loop length. **Right** Loops colored by their canonical cluster.

### E.5 RECOVERY OF THE CANONICAL CLUSTERS

In Table S7 we report how well IGLOO can recover Kelow et al. (2022) clusters based on the IGLOO quantized token, $\hat{t}$. We compare the results with (1) IGLOO with the dihedral angles masked out and only sequence input, and (2) IGLOO with the sequence masked out and only dihedral angles input. IGLOO performs best when dihedral angles are provided, though the dihedral angles can also be obtained through protein structure prediction models. Inference with only sequence input is suitable when encoding large libraries of antibody sequences, where structure prediction would be too computationally intensive. We observe that the performance of the sequence-only IGLOO is very close to the IGLOO model for most loop types except for the H3 and L3 loops.

Table S7: Average IGLOO cluster purity ($\uparrow$) of North et al. (2011) defined clusters of antibody CDRs across SAbDab.

| Loop Type | IGLOO | IGLOO sequence only | IGLOO dihedral angles only |
|---|---|---|---|
| H1 | 0.894 | 0.880 | 0.898 |
| H2 | 0.900 | 0.875 | 0.914 |
| H3 | 0.754 | 0.537 | 0.725 |
| H4 | 0.983 | 0.996 | 0.979 |
| L1 | 0.880 | 0.841 | 0.867 |
| L2 | 0.975 | 0.991 | 0.976 |
| L3 | 0.831 | 0.771 | 0.812 |
| L4 | 0.930 | 0.928 | 0.914 |

## E.6 IGLOO CODEBOOKS

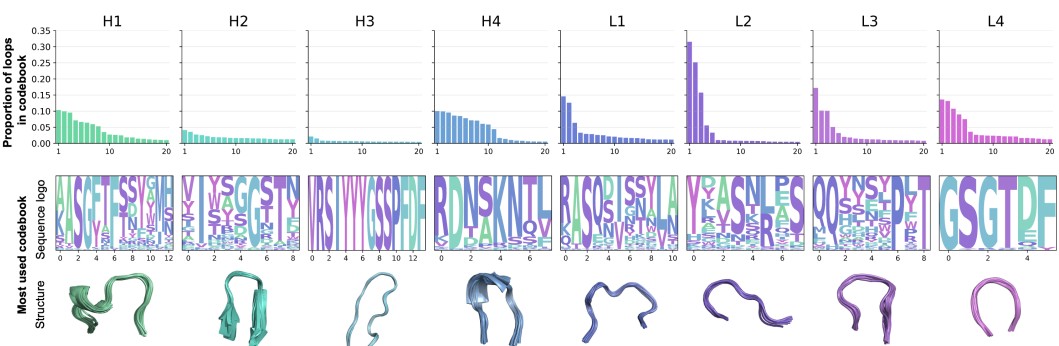

Figure S2: **Top** Top 20 used IGLOO codebooks for each CDR type in SAbDab. **Bottom** Sequence logo and aligned structures of 20 loops for the most used IGLOO codebook for each loop type.

## E.7 PREDICTING BINDING AFFINITY WITH IGLOO TOKENS

In Table S8, we show results of IGLOOALM compared to IGLOOLM in predicting binding affinity for AbBiBench. IGLOOALM additionally includes multimodal residue tokens from IGLOO which encode the dihedral angle backbone of the loop residues. We observe a drop in performance when these tokens are added. This is consistent with the lower performance of protein language models which use Foldseek 3Di tokens (SaProt and ProstT5 in Table 2). In the binding-affinity benchmark, models must distinguish subtle differences among a small set of heavy-chain variants. State-of-the-art structure-prediction models often miss these nuanced conformational changes (Pak et al., 2023; Buel & Walters, 2022), and the resulting errors propagate to protein language models that rely on residue-level structural tokens.

Table S8: Spearman correlation coefficient ($\uparrow$) for binding affinity prediction on AbBiBench for IGLOOLM and IGLOOALM.

| Target | IGLOOLM | IGLOOALM |
|---|---|---|
| 1mlc | 0.616 (0.009) | 0.513 (0.020) |
| 1n8z | 0.675 (0.025) | 0.556 (0.023) |
| 2fjg | 0.713 (0.014) | 0.635 (0.015) |
| 3gbn_h1 | 0.948 (0.004) | 0.929 (0.005) |
| 3gbn_h9 | 0.962 (0.003) | 0.959 (0.002) |
| 4fqi_h1 | 0.921 (0.001) | 0.886 (0.001) |
| 4fqi_h3 | 0.971 (0.001) | 0.967 (0.001) |
| aayl49 | 0.625 (0.010) | 0.552 (0.007) |
| aayl49_ML | 0.531 (0.007) | 0.493 (0.007) |
| aayl51 | 0.579 (0.011) | 0.545 (0.014) |

## E.8 CONTROLLABLE SAMPLING OF ANTIBODY LOOPS

In Figure S3 we present additional examples of IGLOOALM sampled sequences and their predicted structures. We present examples where the sampled loops have on average at most 60% sequence identity with the original loop.

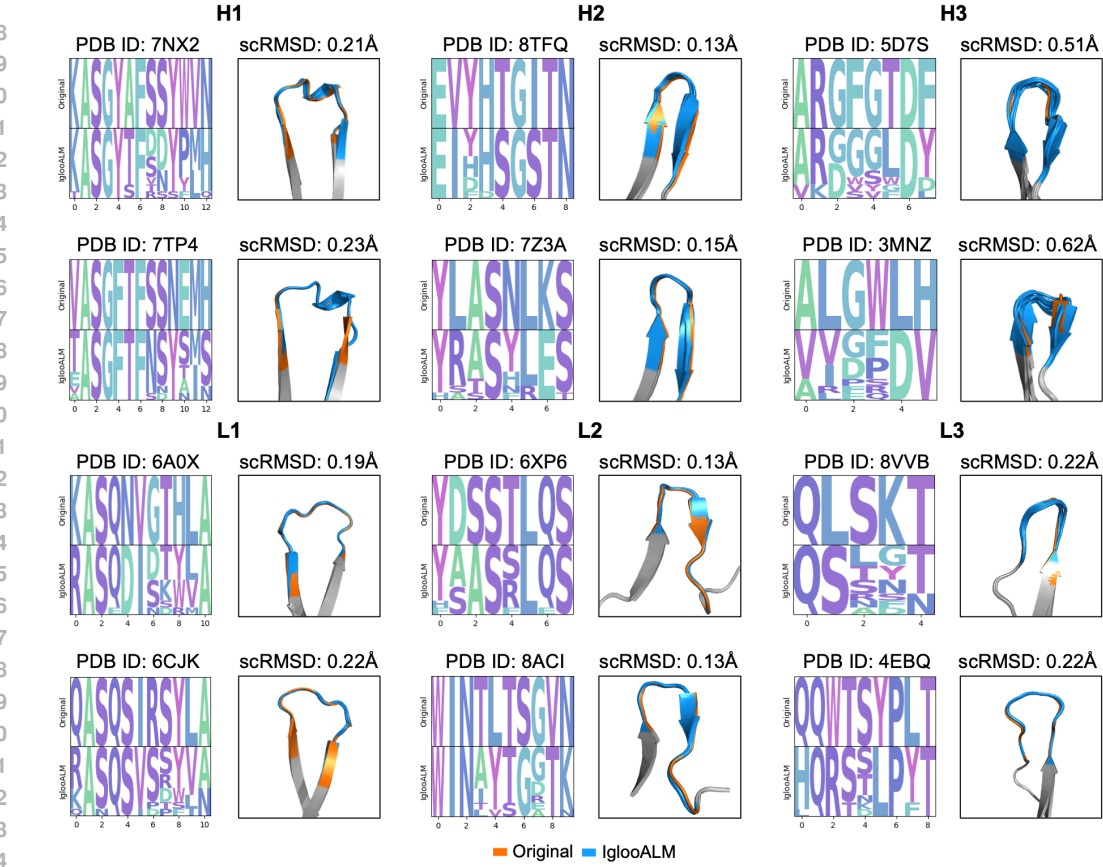

Figure S3: Sequence logo of original and ten IGLOOALM sampled sequences of CDR loop regions at $\lambda = 0.5$ and predicted structures.

