# OpenReview forum: "Tokenizing Loops of Antibodies"
_ICLR.cc/2026/Conference — ICLR 2026 Conference Withdrawn Submission_

### Official Review · Reviewer_eXQi · 2025-10-28

**Soundness:** 2
**Presentation:** 3
**Contribution:** 3
**Rating:** 4
**Confidence:** 2

**Summary:**

This paper proposes a new multimodal antibody loop tokenizer that encodes backbone dihedral angles and sequence.  The tokenizer mainly uses a contrative learning objective to enhance discrimination ability for loop similiarity. The exprimental results showcase the methods achieve SOTA preformance.

**Strengths:**

- The presentation is good and the paper is reay to read and understand.
- The CDRs is important and conducting the research on the CDRs is meaningful.
- The motivation to consider the structure and sequence simutaneously sounds reasonable and makes sence.
- The experimental resutls suggest the model is good.

**Weaknesses:**

- Although the paper is easy to read, the nolvety of this method is limited. It doesn't introdue any insignful training technique and model archtecture design. It seems to accumuate some tricks (mask seq, contrastive learning, codebook) to achieve the good performance on the loop representation. Therefore, for the methods itself, it cannot be said bring a significant contribution for the community.
- The experimental resutls suggest the proposed methods can be significantly consistantly outperforms the baseline.

minor: - line 58 write the full name of the abbreviation MD.
-  Eq (3) $\ell_{AA}$ needs description.

**Questions:**

what if the model learns 6 loopers instead of 4 loopers for CDRs.
For 4.4, can the diffab and dyMEAN be the baselines?

[1] End-to-End Full-Atom Antibody Design

---

> ### Author Response · Authors · 2025-11-19
>
> *1. Although the paper is easy to read, the nolvety of this method is limited. It doesn't introdue any insignful training technique and model archtecture design. It seems to accumuate some tricks (mask seq, contrastive learning, codebook) to achieve the good performance on the loop representation. Therefore, for the methods itself, it cannot be said bring a significant contribution for the community.*
>
> While Igloo leverages established components such as masking and contrastive learning, its novelty lies in how these are adapted to the antibody loop representation problem. Igloo introduces a dihedral distance–based contrastive objective for loop-level structural alignment, a multimodal tokenizer combining sequence and backbone geometry, and a method to integrate loop tokens into existing protein language models.
>
> Advancements for the antibody field:
> * **Structure-aware antibody design**: Igloo loop tokens can be readily incorporated into generative antibody models to guide sequence design toward desired backbone conformations (Sec. 4.4).
> * **Transferable structural priors**: The learned loop tokens provide interpretable embeddings and can serve as a drop in replacement for existing uses of the canonical clusters. For example, Igloo can inform affinity maturation, binding affinity prediction, and prioritization of large antibody libraries.
> * **Data-efficient modeling**: Because Igloo captures shared geometry across loop types, rather than existing approaches which model the CDR loops separately, it can improve learning in low-data regimes (e.g., rare CDR classes, nanobodies). Compared to existing canonical clusters, Igloo achieves coverage for all CDR H3 loops.
>
> Advancements for the machine learning field:
> * **Substructure-level representation learning**: Igloo demonstrates how to tokenize biomolecular subdomains (loops, motifs) as discrete units, bridging residue-level and protein-level representations.
> * **Geometry-aware contrastive learning**: The dihedral-angle based contrastive objective provides a rotation-invariant and chemically grounded alternative to RMSD-based training for structural encoders.
> * **Scalable multimodal token integration**: The integration of learned loop tokens into language models (IglooLM, IglooALM) represents a new paradigm for modular structure conditioning for antibody language models.
>
> *2. The experimental resutls suggest the proposed methods can be significantly consistantly outperforms the baseline.*
>
> We would like to highlight that IGLOO achieves improvements across multiple independent tasks.
>
> In Sec. 4.1, Igloo is better able to **retrieve loops with similar dihedral angles** than recently published Amino Aseed [1] (+6.1% for CDR3) and FoldSeek [2] (+11.0% for CDR3).
>
> [1] Yuan, X., Wang, Z., Collins, M., & Rangwala, H. (2025). Protein Structure Tokenization: Benchmarking and New Recipe. In Proceedings of the 42nd International Conference on Machine Learning (ICML 2025).
>
> [2] Van Kempen, M., Kim, S. S., Tumescheit, C., Mirdita, M., Lee, J., Gilchrist, C. L., ... & Steinegger, M. (2024). Fast and accurate protein structure search with Foldseek. Nature biotechnology, 42(2), 243-246.
>
> In Sec. 4.3, we show **advancements in predicting binding affinity**. Notably IglooLM performs on average better than ESM-2 (3B) despite having 7x less parameters. Compared to other multimodal language models SaProt (1.3B) and ProstT5 (3B), IglooLM achieves better performance on 9/10 and 10/10 targets respectively.
>
> In Sec. 4.4 we show when **sampling antibody loops with Igloo we achieve better structural consistency** for a given sequence identity compared to recently published inverse folding models IgBert, AntiFold and AbMPNN. Igloo outperforms AbMPNN for 9 out of 13 sequence-identity bins, Igloo generates loops with higher structural consistency (lower self-consistency RMSD) to the original loop, while maintaining comparable sequence diversity.
>
> In *newly added* Sec. 4.5, we demonstrate that **Igloo enables zero-shot, highly scalable hit-to-lead optimization** by prioritizing experimentally-validated CDR H3 loop variants for the HER2 target. Sequences sharing the same Igloo token as the seed loop exhibit 55.3\% experimentally validated binders, a $1.9\times$ enrichment over the baseline library.
>
> *3. minor: - line 58 write the full name of the abbreviation MD.*
>
> We appreciate the reviewer in raising this. We have addressed this in our revised submission on L58.
>
>
> *4. Eq (3)* $\ell_{AA}$ *needs description*
>
> We appreciate the reviewer in raising this. We have addressed this in our revised submission on L197.
> $\ell_{\text{AA}, i} = - \sum_{a=1}^{20} y_{i,a} \log \hat{p}_{i,a}$

---

> > ### Author Response · Authors · 2025-11-27
> >
> > Dear Reviewer,
> >
> > We hope you are well! We would be grateful if you could confirm whether our rebuttal and additional experiments have addressed your original concerns. We deeply value your feedback and would appreciate if you could share your thoughts as soon as possible.
> >
> > Thank you again for your invaluable time and constructive feedback.

---

### Official Review · Reviewer_4kVJ · 2025-10-30

**Soundness:** 3
**Presentation:** 3
**Contribution:** 2
**Rating:** 4
**Confidence:** 4

**Summary:**

The paper introduces IGLOO, a novel multimodal tokenizer designed for the CDRs of antibodies. The key contribution is the use of a higher-level tokenization of an entire CDR loop, rather than residue-level tokenization in common protein language models. IGLOO is trained to create continuous and discrete (via a codebook) representation of the CDR loop. The training incorporates three objectives: multimodal masking for reconstruction, codebook learning for discrete tokenization, and a contrastive learning objective based on dihedral angle distance, which aims to map structurally similar loops close in the latent space. The authors demonstrate the utility of these learned tokens across three tasks in antibody engineering: (1) retrieving structurally similar loops from large databases, (2) binding affinity predictions by incorporating the loop tokens, and (3) enabling controllable sampling of diverse yet structurally consistent antibody loops for design.

**Strengths:**

1. The idea of tokenizing at the level of CDR loops is suitable for CDR level tasks. The contrastive learning objective based on dihedral angle distance is novel for CDR representation learning, as it captures the chirality and geometry of protein backbones more effectively than standard RMSD.

2. The extensive experiments have been done on three related tasks.

3. The paper is well-written and organized. The method is clearly explained.

**Weaknesses:**

1. The author used a structure prediction tool to fold the antibody sequences from OAS. Currently, the accuracy of a folding model on the CDR H3 is usually > 2 Å. Is it okay to use such data for training? Thus, the use of the precision for RMSD < 1 Å may be questionable. What is the quality of the VQVAE reconstruction?

2. The improvement of the performance of binding affinity prediction is marginal.

3. The experiment on the antibody structure clustering is not compared to other methods.

4. In the contrastive loss, I guess that it may be possible that CDRs with different lengths have similar structures and functions.

**Questions:**

See Weaknesses

---

> ### Author Response · Authors · 2025-11-19
>
> We appreciate the reviewer for higlighting the clarity and organization of the paper, and the extensive experiments conducted. Below we address each of your concerns.
>
> *1. The author used a structure prediction tool to fold the antibody sequences from OAS. Currently, the accuracy of a folding model on the CDR H3 is usually > 2 Å. Is it okay to use such data for training? Thus, the use of the precision for RMSD < 1 Å may be questionable. What is the quality of the VQVAE reconstruction?*
>
> We thank the reviewer for raising this important point. We fully acknowledge that current structure prediction tools can exhibit deviations of >2 Å on CDR H3 loops. Predicted structures from OAS were used *for training only* to increase the diversity of the relatively limited ~15,000 experimentally resolved antibody structures available in SAbDab. We would like to highlight that for the **evaluation of precision for RMSD < 1Å these loops were performed exclusively on experimentally resolved antibody structures from SAbDab**. We clarify this in L327, 343. This approach allows Igloo to learn from a broader structural prior while preserving the benchmark validity of Table 1 against experimentally verified data. For the reconstruction quality, we report the amino acid recovery and mean absolute circular error for each of the dihedral angles in Table S4 on page 19.
>
> *2. The improvement of the performance of binding affinity prediction is marginal.*
>
> We agree that the absolute improvement in binding affinity prediction is moderate. **Notably IglooLM achieves on average +1.7% improvement to ESM-2 (3B) but is a 7x smaller model.** Because these datasets differ by only a few point mutations, even small gains in Spearman correlation reflect non-trivial improvements in capturing structural determinants of binding affinity through Igloo’s loop tokens. We highlight the per-parameter efficiency in Table A of our official comment above, and added parameter counts to Table 2 of the paper.
>
> *3. The experiment on the antibody structure clustering is not compared to other methods.*
>
> The goal of the clustering analysis in Sec. 4.2 was not designed as a benchmarking task, but rather to characterize the structural organization learned by Igloo. The purpose of this section is therefore exploratory rather than comparative and to reflect meaningful structural organization of the Igloo latent space. Igloo recovers canonical antibody clusters with an average purity of 89.3% for canonical clusters, confirming that its learned representations reflect meaningful structural organization. Unlike canonical clustering approaches, where 76.3% of CDR H3 loops are not assigned to a canonical cluster, Igloo learns representations for all CDR H3 loops, overcoming this coverage limitation.
>
> *4. In the contrastive loss, I guess that it may be possible that CDRs with different lengths have similar structures and functions.*
>
> We agree that CDR loops of different lengths can sometimes adopt similar backbone conformations and even share functional roles. To explore this, we had already trained a variant of Igloo where positive pairs were defined between loops of different lengths using dynamic time warping alignment of the loop backbones, following Nowak et al. (2016). *This analysis was presented in Appendix D (Algorithm 1 and Table S3) in our original submission*. We observed that incorporating cross-length positive pairs leads to comparable Precision@k performance to the main model, with a slight reduction in accuracy on some loop types. For stability and interpretability, we therefore focused our main model on same-length contrastive pairs. Though, our implementation of Igloo with comparisons across different length loops is available in the codebase we shared.

---

> > ### Author Response · Authors · 2025-11-27
> >
> > Dear Reviewer,
> >
> > We hope you are well! We would be grateful if you could confirm whether our rebuttal and additional experiments have addressed your original concerns. We deeply value your feedback and would appreciate if you could share your thoughts as soon as possible.
> >
> > Thank you again for your invaluable time and constructive feedback.

---

### Official Review · Reviewer_9Yoh · 2025-10-30

**Soundness:** 2
**Presentation:** 2
**Contribution:** 2
**Rating:** 2
**Confidence:** 3

**Summary:**

The paper proposes a multimodal algorithm that encodes antibody loop regions by integrating both amino acid sequences and backbone dihedral angles to generate unified token representations. IGLOO uses a transformer-based architecture built on ESM-2, trained through three self-supervised objectives: multimodal masking with reconstruction of sequence and angle features, contrastive learning based on dihedral angle distance to cluster structurally similar loops closer in latent space, and codebook learning for transforming continuous embeddings into discrete tokens that capture canonical conformations. These learned tokens are applied to two structure-aware antibody language models, IGLOOLM and IGLOOALM, enabling improved retrieval of similar loop structures, higher performance in binding affinity prediction, and controllable generation of structurally consistent antibody loops, thereby advancing multimodal foundation modeling for rational antibody design.

**Strengths:**

1 IGLOO introduces a novel multimodal tokenization framework that jointly encodes antibody loop sequences and backbone dihedral angles, enabling unified structural–functional representation beyond traditional canonical clustering methods.

2 By embedding IGLOO tokens into antibody language models (IGLOOLM and IGLOOALM), the approach provides structure-aware contextual understanding, improving affinity prediction and controllable loop generation.

**Weaknesses:**

1 Although this work makes multiple algorithm innovations, the results seem to be limited compared to its baselines. This makes me have doubts about the usability of this work.

2 IGLOO represents structures only through backbone dihedral angles, omitting side-chain conformations, hydrogen-bond networks, and atomic-level interactions. While this simplifies computation, it may fail to capture fine-grained spatial details that affect loop conformation and binding affinity. Please explain why you didn’t incorporate all-atom geometric features or graph-based structural encoders to better model spatial constraints, and add relative baselines.

3 The model defines positive and negative loop pairs using fixed dihedral distance thresholds (D < 0.1 and D > 0.47). These hard boundaries could lead to poor representation of loops with intermediate similarity. Similarly, please explain why you didn’t consider a more continuous similarity weighting or soft contrastive loss.

4 IGLOO is trained mainly on SAbDab and Ibex-predicted antibodies, which reflect known scaffolds and conformations. Its generalizability to noncanonical or novel antibody frameworks remains untested. Broader training with non-antibody loop structures or cross-species datasets could improve robustness and transfer capability.

5 The paper compares IGLOO to large protein language models like ESM-2 (3B) and multimodal models such as ProstT5 and SaProt, but these baselines differ significantly in scale and objectives, making direct efficiency comparisons less clear. Future studies could include normalized metrics (e.g., per-parameter efficiency) or scaling analyses at similar model sizes.

6 Benchmark tasks focus on structure retrieval and affinity prediction but lack direct comparisons on design-oriented tasks against models such as ProteinMPNN or AbMPNN. Evaluating loop-generation quality or binding-site prediction would more fully demonstrate IGLOO’s design advantages.

7 Current metrics—RMSD and dihedral distance (D)—capture geometric similarity but not functional performance, such as binding energy or antigen-interaction context. Incorporating binding free energy (ΔΔG) and interface contact consistency metrics could assess biological relevance more effectively.

**Questions:**

Same as weakness.

---

> ### Author Response · Authors · 2025-11-19
>
> We appreciate your recognition of the novelty of our work. Below we address each of your concerns.
>
> **1.** We present **four main applications of Igloo**. In Sec. 4.1, Igloo is able to **retrieve loops with similar backbones**, outperforming Amino Aseed (+6.1% for CDR H3) and FoldSeek (+11.0% for CDR H3). In Sec. 4.3, **IglooLM for binding affinity prediction** performs better than ESM-2 (3B) despite having 7x less parameters. Compared to multimodal language models SaProt (1.3B) and ProstT5 (3B), IglooLM achieves better performance on 9/10 and 10/10 targets respectively. In Sec. 4.4, **sampling antibody loops** with Igloo achieves better structural consistency for a given sequence identity, and outperforms AbMPNN for 9 out of 13 sequence-identity bins. We also added Sec. 4.5, which shows an impactful and direct application of Igloo, where it can **rapidly and scalably prioritize functional antibody variants from large mutagenesis libraries**.
>
> **2.** We appreciate the reviewer’s insightful comment, which the paper already discussed in Sec. 5 L538-539. The goal of this work was to introduce a tokenizer that captures the backbone conformational diversity of antibody loops. We intentionally focused on backbone dihedral angles because they compactly represent loop geometry in a rotation and translation-invariant manner and directly correspond to canonical definitions of antibody loop clusters, which are currently used by the antibody research community. We agree that including finer-grained spatial and interaction features could further improve modeling of spatial constraints. As for relevant baselines, we included SOTA protein backbone tokenisers, including recently published Amino Aseed [1].
>
> [1] Yuan, X., Wang, Z., Collins, M., & Rangwala, H. (2025). Protein Structure Tokenization: Benchmarking and New Recipe. In Proceedings of the 42nd International Conference on Machine Learning (ICML 2025).
>
> **3.** We adopted the fixed dihedral thresholds (D < 0.1 and D > 0.47) to maintain consistency with canonical definitions of antibody loop conformations (North et al., 2011; Kelow et al., 2022), ensuring interpretability of learned clusters. To avoid artifacts from examples near the threshold, pairs with intermediate D values were excluded from training, effectively creating a soft boundary. As shown in Table S3, this soft boundary improved retrieval precision. We agree that soft or continuous similarity weighting could further refine the latent structure and plan to explore this direction in future work.
>
> **4.** We intentionally focus on antibody CDR loops, which are the most conformationally diverse and functionally critical regions. Modeling these highly variable loops is challenging and has **immediate pharmaceutical relevance**. While Igloo is trained on SAbDab and Ibex antibodies to ensure data quality and coverage, we agree that extending to non-antibody and cross-species loops is a valuable future direction.
>
> **5.** Our intention was to highlight that IglooLM (420M parameters) **achieves performance exceeding models that are up to 7x larger** such as the protein language model ESM-2 (3B), and multimodal protein language models ProstT5 (3B), SaProt (1.3B). For more comparable models ESM-C (300M), IglooLM outperforms for 9/10 targets and IgBert (420M) outperforms for 8/10 targets. We include in Table A in our official comment above the per-parameter efficiency, showing that IglooLM has a much higher per-parameter efficiency than ESM-2 (3B), ProstT5 and SaProt.
>
> **6.** We would like to clarify that Igloo’s loop-design capability has already been directly benchmarked against AbMPNN in Figure 4. **Igloo outperforms AbMPNN for 9 out of 13 sequence-identity bins**, Igloo generates loops with higher structural consistency (lower self-consistency RMSD) to the original loop, while maintaining comparable sequence diversity. Regarding binding-site prediction, we note that Igloo is explicitly focused on modeling CDR loop conformations, which is essential for antigen binding. As such, binding-site identification falls outside the scope of this work. Our aim is to provide a loop-level structural tokenizer that can be integrated into generative or predictive frameworks for antibody design, rather than to predict binding epitopes directly.
>
> **7.** We fully agree that the suggested metrics provide valuable measures of biological relevance. However, we aimed to establish a geometric representation and tokenization framework for antibody loops, RMSD and dihedral distance (D) were therefore chosen as appropriate, well-established geometric metrics directly aligned with canonical loop definitions and the structural objectives of the model. Because Igloo tokens can be integrated into downstream predictive or generative models (as demonstrated in the affinity prediction and loop generation tasks), future extensions can readily incorporate ΔΔG estimation or interface-level metrics to directly evaluate functional performance.

---

> > ### Author Response · Authors · 2025-11-27
> >
> > Dear Reviewer,
> >
> > We hope you are well! We would be grateful if you could confirm whether our rebuttal and additional experiments have addressed your original concerns. We deeply value your feedback and would appreciate if you could share your thoughts as soon as possible.
> >
> > Thank you again for your invaluable time and constructive feedback.

---

> ### Comment · Reviewer_9Yoh · 2025-11-28
>
> Thank you for your hard work, but I'm not quite convinced. I'll keep my original score.

---

### Author Response · Authors · 2025-11-19

We thank all reviewers for their constructive and thoughtful feedback. We appreciate the acknowledgements of the paper’s strengths including the *novelty of the multimodal tokenization framework* from 9Yoh and 4kVj, the *quality and clarity* of the paper and method from 4kVj, and the *strong experimental results* recognized by eXQi.

The comments have also helped us strengthen the paper and clarify its contributions. The reviewers raised several key points regarding (1) contribution and scope of Igloo, (2) applications of Igloo, (3) the interpretability and significance of experimental results, and (4) requests for additional clarification or analysis. We summarize below how the revised version and our responses address these points.

**(1) Contribution and Scope.** Igloo establishes the *first multimodal tokenizer for antibody loops*, providing a unified geometric and sequence-based representation that can be directly incorporated into protein language models. This loop-level formulation introduces a new representational scale for antibodies, bridging residue-level structural data with higher-level functional modeling. The approach enables improved structure retrieval, loop generation, and affinity prediction, while also offering an interpretable structural vocabulary for antibody engineering.

**(2) Zero-shot prioritization of binders from large libraries.** Reviewers expressed concerns about the applications of Igloo. In our revisions, we demonstrate that Igloo can rapidly and scalably prioritize functional antibody variants from large mutagenesis libraries, achieving up to a **$\mathbf{1.9\times}$ enrichment of experimentally-validated HER2 binders in a zero-shot setting**. See the added Sec. 4.5 for full details.

**(3) Per-Parameter Efficiency of Igloo.** As raised by reviewer 9Yoh we now include a parameter-efficiency analysis (Table A) demonstrating that representations from IglooLM achieves on average better performance with a +1.7% improvement compared to ESM2 (3B) on binding affinity prediction and much better per-parameter efficiency than models with 7x more parameters.

**Table A:** Per-parameter efficiency for average spearman correlation for binding affinity prediction on AbBiBench.
| **Model** | **ESM2** | **ProstT5** | **SaProt** | **IglooLM** |
|:--|:--:|:--:|:--:|:--:|
| Average \\(\\rho\\) for 10 AbBiBench targets (\\(\\downarrow\\)) | 0.7413 | 0.4421 | 0.7233 | **0.7541** |
| Parameters (B) (\\(\\downarrow\\)) | 3.00 | 3.00 | 1.30 | 0.42 |
| Avg. \\(\\rho\\) / Parameters (B) (\\(\\uparrow\\)) | 0.2471 | 0.1474 | 0.5564 | **1.7955** |

**(4) Clarifications and Additional Analyses**
* All precision evaluations in Table 1 are performed on *experimentally resolved structures* to ensure benchmark validity.
* Section 4.2 has been clarified as an exploratory analysis. Igloo recovers canonical antibody clusters with an average purity of 89.3% for canonical clusters, confirming that its learned representations reflect meaningful structural organization. Unlike canonical clustering approaches, where 76.3% of CDR3 loops are not assigned to a canonical cluster, Igloo learns representations for all CDR3 loops, overcoming this coverage limitation.
* Appendix D indeed already includes analyses defining cross-length positive pairs using dynamic time warping, demonstrating comparable performance as suggested by reviewer 4kVj.

We believe these clarifications and analyses demonstrate that Igloo establishes a multimodal representation framework for antibody loops that advances widely used antibody canonical clusters, and multimodal antibody language modeling. Through our extensive experiments, we also show that **Igloo has direct and impactful applications for antibody design**.

---

### Comment · Area_Chair_a3eq · 2025-11-28
**Please check the rebuttal and give feedback**

Dear Reviewers,

Thanks for your effort in reviewing the manuscript. Now the authors have provided the rebuttal and it's highly recommended to take a look and give your feedback. Thanks.

AC.

---

### Note · Authors · 2025-12-02

I have read and agree with the venue's withdrawal policy on behalf of myself and my co-authors.